# Deconvolution of Buparlisib's mechanism of action defines specific PI3K and tubulin inhibitors for therapeutic intervention

Thomas Bohnacker[1], Andrea E. Prota[2,*], Florent Beaufils[1,*,†], John E. Burke[3,*], Anna Melone[1], Alison J. Inglis[4], Denise Rageot[1], Alexander M. Sele[1], Vladimir Cmiljanovic[1,†], Natasa Cmiljanovic[1,†], Katja Bargsten[2,†], Amol Aher[5], Anna Akhmanova[5], J. Fernando Díaz[6], Doriano Fabbro[7], Marketa Zvelebil[8], Roger L. Williams[4], Michel O. Steinmetz[2] & Matthias P. Wymann[1]

BKM120 (Buparlisib) is one of the most advanced phosphoinositide 3-kinase (PI3K) inhibitors for the treatment of cancer, but it interferes as an off-target effect with microtubule polymerization. Here, we developed two chemical derivatives that differ from BKM120 by only one atom. We show that these minute changes separate the dual activity of BKM120 into discrete PI3K and tubulin inhibitors. Analysis of the compounds cellular growth arrest phenotypes and microtubule dynamics suggest that the antiproliferative activity of BKM120 is mainly due to microtubule-dependent cytotoxicity rather than through inhibition of PI3K. Crystal structures of BKM120 and derivatives in complex with tubulin and PI3K provide insights into the selective mode of action of this class of drugs. Our results raise concerns over BKM120's generally accepted mode of action, and provide a unique mechanistic basis for next-generation PI3K inhibitors with improved safety profiles and flexibility for use in combination therapies.

[1] Department of Biomedicine, University of Basel, 4058 Basel, Switzerland. [2] Laboratory of Biomolecular Research, Department of Biology and Chemistry, Paul Scherrer Institut, 5232 Villigen, Switzerland. [3] Department of Biochemistry and Microbiology, University of Victoria, Victoria, British Columbia BC V8W 2Y2, Canada. [4] MRC Laboratory of Molecular Biology, Cambridge CB2 2QH, UK. [5] Cell Biology, Faculty of Science, Utrecht University, 3584 CH Utrecht, The Netherlands. [6] CIB Centro de Investigaciones Biológicas, 28040 Madrid, Spain. [7] PIQUR Therapeutics AG, 4057 Basel, Switzerland. [8] The Institute of Cancer Research, London SW3 6JB, UK. * These authors contributed equally to this work. † Present address(es): PIQUR Therapeutics AG, 4057 Basel, Switzerland (F.B. or V.C. or N.C.); Institute of Biochemistry, University of Zürich, 8057 Zürich, Switzerland (K.B.). Correspondence and requests for materials should be addressed to M.P.W. (email: Matthias.Wymann@UniBas.CH).

Phosphoinositide 3-kinases (PI3Ks) promote cell growth, survival, division and motility, and signal through protein kinase B (PKB/Akt) and the mammalian target of rapamycin (mTOR) to control cellular growth and proliferation. The PI3K/PKB/mTOR pathway is frequently upregulated in a wide range of tumours, and is therefore considered a valuable drug target in cancer therapy. Specific PI3K inhibitors typically display a cytostatic action, and cause cell cycle arrest in the G1/S phase[1]. To date, an impressive number of PI3K and mTOR inhibitors targeting the ATP-binding pocket of PI3K and PI3K-related kinases (PIKK) are in clinical trials[2]. Among those, NVP-BKM120 (BKM120) is one of the clinically most advanced pan-PI3K inhibitors, as it is enlisted in more than 80 clinical studies as a single drug or in combination therapies[3–6].

In contrast to targeted therapy by PI3K inhibition, classical chemotherapeutic agents perturbing microtubule dynamics cause mitotic arrest and cytotoxicity[7]. BKM120 is a potent PI3K inhibitor, but has been reported to also act as a microtubule-destabilizing agent (MDA). In spite of this finding, it has been claimed that there is a therapeutic window where BKM120 targets PI3K selectively without any interference with microtubule polymerization[8]. This conclusion was reached, however, without a defined binding site for BKM120 on tubulin and an established molecular mechanism of action of microtubule disruption, and was additionally based on an incomplete understanding of the interaction of BKM120 with PI3K.

BKM120 can thus principally act as a PI3K inhibitor or as a chemotherapeutic agent and its dominant antitumorigenic action needs to be carefully evaluated to avoid misinterpretation of preclinical and clinical data. The convoluted mode of action of BKM120 also compromises the evaluation of biomarkers, and a rational set-up and evaluation of drug combination therapies. If one attributes BKM120's antiproliferative action to PI3K inhibition, elimination of its MDA activity should be beneficial, as MDAs often display severe toxic side effects upon prolonged therapy[9]. A comprehensive elucidation of the mode of action of BKM120 is therefore crucial to improve the rationale of clinical studies involving buparlisib drug combinations, and will also contribute to the development of next-generation PI3K inhibitors.

Here, a combination of chemical, structural and biological approaches was used to dissect the above-mentioned ambiguities and provide a detailed understanding of the molecular interactions of BKM120 with tubulin and PI3K.

## Results

**Derivatives of BKM120 deconvolute its MDA and PI3K activity**. As a first step we aimed to separate the two biological activities of BKM120 and to produce related chemical derivatives that specifically target either PI3K or tubulin. An important part of the approach was to retain the drug-like properties of the parental compound. An exchange of the core pyrimidine with pyridine produced microtubule targeting drug 147 (MTD147), a compound with pronounced MDA activity and minimal PI3K inhibition. Replacing the BKM120 core with triazine yielded PQR309, which excelled as a potent pan-PI3K inhibitor with no detectable MDA activity. PI3K activity was monitored by phosphorylation of PKB/Akt, and phosphorylation of Histone H3 was used as an indicator of mitotic arrest (Fig. 1a,b, Supplementary Fig. 1a,b and Supplementary Fig. 1i–m).

In five reference cancer cell lines (A2058, BT-549, SKOV-3, U87-MG and HCT116) the established MDAs nocodazole and colchicine, as well as MTD147 and BKM120, all triggered Histone H3 phosphorylation (Supplementary Fig. 1b and Supplementary Fig. 1i–m), nuclear DNA condensation (Supplementary Fig. 1c

and Supplementary Fig. 1i–m) and the accumulation of cells in G2/M or sub-G1 fractions. These events are clear evidence of mitotic arrest or entry into apoptosis (Supplementary Fig. 1e–h). In contrast, PQR309 and the structurally unrelated pan-PI3K inhibitor GDC0941 did not display apoptotic activities and arrested cells in the G1/S phase. Although BKM120 and PQR309 achieved half-maximal growth inhibition at indistinguishable concentrations at approximately 1 μM (Supplementary Fig. 1d and Supplementary Fig. 1i–m), they clearly showed a different mode of action. Combined, the above assays indicate that the MDA activity of BKM120 dominates its biological action.

An enlarged 44-cell line panel monitoring cell viability and proliferation confirmed the above reference cell line values and was exploited to collect $IC_{50}$-independent compound characteristics: the comparison of cell line drug sensitivity profiles revealed that PQR309 had highest similarity to the PI3K inhibitor reference compounds GDC0980 and GDC0941, while BKM120 and MTD147 deviated from PI3K-inhibitor sensitivity patterns (Fig. 1c and Supplementary Tables 1 and 2). This divergence was reinforced by Hill slope analysis of dose–response curves for cell viability, which has been used before to distinguish distinct drug classes[10]. Hill slopes close to –1 were obtained for PQR309 and the two PI3K-inhibitor controls. BKM120 and MTD147, however, displayed Hill slope values of $< -2$ (Fig. 1d,e and Supplementary Table 3), which were statistically indistinguishable from Hill slopes obtained with nocodazole and colchicine (Supplementary Fig. 1n).

Cell line-specific cross-correlations of Hill slopes for compound pairs revealed a high similarity of PQR309 with PI3K inhibitor reference compounds, while BKM120 and MTD147 displayed an increased variance (Fig. 1f and Supplementary Fig. 1o). Furthermore, least-square penalty score calculations for Hill slopes were close to baseline for pairs of PQR309, GDC0980 and GDC0941, while BKM120 and MTD147 penalties exceeded values of 100 when cross-correlated to any PI3K inhibitor (Fig. 1g). Finally, and consistent with the cell line panel analysis, BKM120 and MTD147 elevated the number of cells positive for phosphorylated Histone H3 in all tested cell lines, whereas PQR309 reduced the number of phospho-Histone H3-positive cells in 38 out of 39 cases (Fig. 1h).

The data described above demonstrate that minimal changes in the pyrimidine core of BKM120 allow a separation of its PI3K and MDA activities into discrete BKM120 derivatives. The matching cell cycle arrest phenotypes of BKM120 and MTD147 indicate that BKM120 prevents proliferation by a mitotic block rather than through inhibition of PI3K.

**BKM120 modulates microtubule dynamics *in vitro* and in cells**. To elucidate whether the mitotic arrest is mediated by a direct perturbation of microtubule dynamics, we analysed the effects of the drugs in an *in vitro* microtubule reconstitution assay and in stably transfected HeLa cells, both using GFP-tagged EB3 as a microtubule plus end marker. We found that MTD147 ($\geq 0.5$ μM) and BKM120 ($\geq 1$ μM), but not PQR309 (5 μM), attenuated microtubule growth rates and increased catastrophe frequency *in vitro* (Fig. 2a–c) and in cells (Fig. 2d–f). As our results match data previously obtained for MDAs such as colchicine[11], we conclude that BKM120 and MTD147 directly perturb microtubule dynamics, and that tubulin binding is the cause of the antiproliferative action of BKM120 and MTD147.

**BKM120-binding site and orientation in αβ-tubulin**. To provide insights into the mechanism of action of BKM120 and MTD147, crystal structures of both compounds bound to tubulin in a complex composed of two αβ-tubulin heterodimers ($T_2$), the

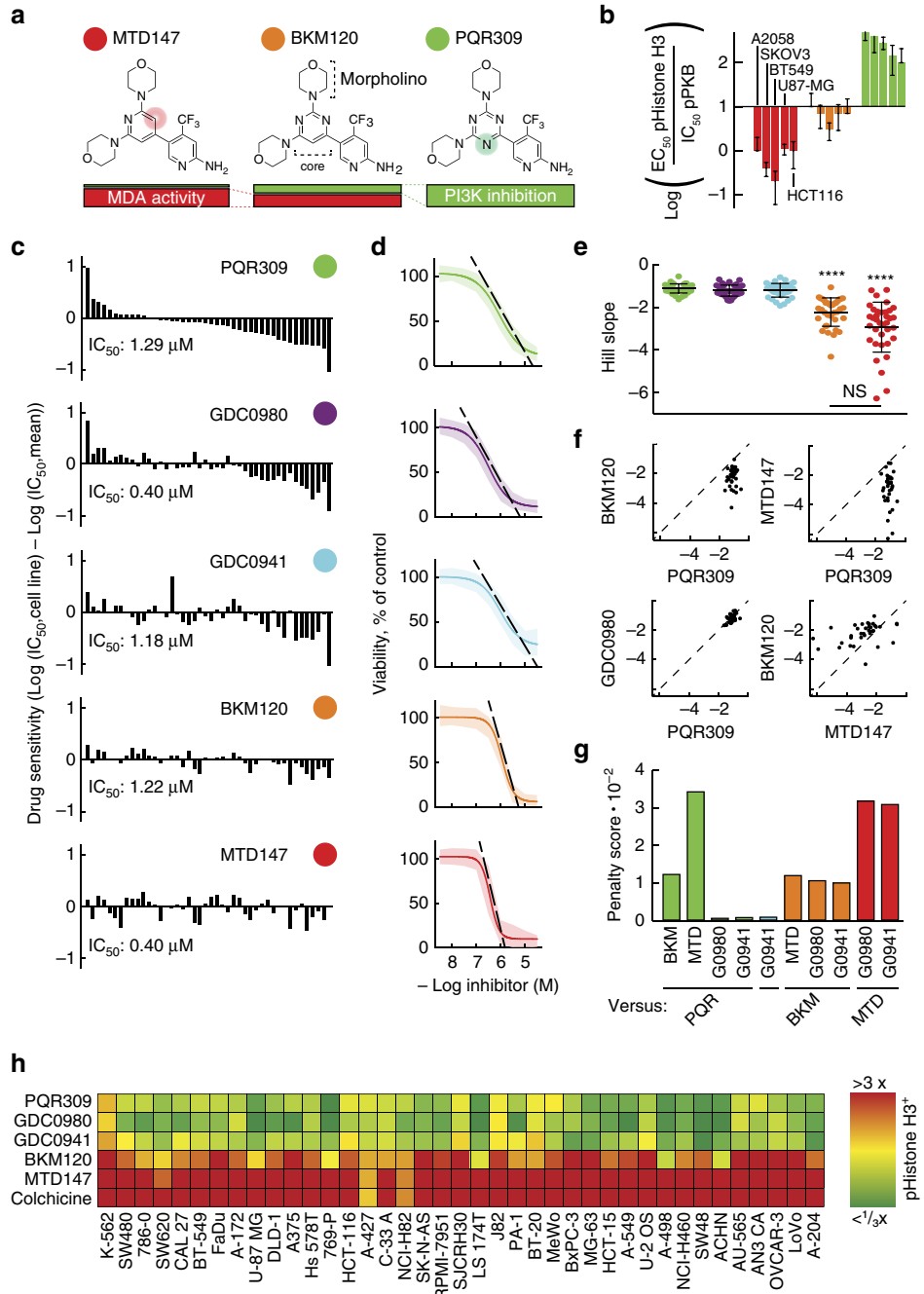

**Figure 1 | Splitting BKM120's inherent biological activities.** (**a**) Exchange of BKM120's pyrimidine core with pyridine yields MTD147; a triazine core is present in PQR309. Rectangles below chemical formulas schematically indicate microtubule-destabilizing agent (MDA, red) and PI3K (green) inhibitor activities. (**b**) Ratios of cellular MDA activity ($EC_{50}$ pHistone H3) over PI3K inhibition ($IC_{50}$ pPKB; compounds colour-coded as in **a**) were determined for the indicated cell lines ($EC_{50}$ of phospho-Histone H3 for PQR309 was set to 20 μM due to the absence of response; for values see Supplementary Materials: Supplementary Fig. 1a,b and Supplementary Fig. 1i–m). PI3K inhibition was assessed by phosphorylation of Akt/PKB (Supplementary Fig. 1a), MDA activity by high-content microscopy detecting phospho-Histone H3 (Supplementary Fig. 1b) and nuclear DNA condensation (Supplementary Fig. 1c), and was correlated with proliferation (Supplementary Fig. 1d). (**c**) Drug sensitivity of 44 cell lines exposed to indicated compounds. Individual $IC_{50}$s of cell line growth were related to the mean $IC_{50}$ of all cells lines, and cell lines were sorted by lowest to highest sensitivity for PQR309 from left to right (for values and cell lines see Supplementary Tables 1,2; mean of duplicate experiments of a 9-point serial drug dilution for each cell line). (**d**) Dose–response curves for cell viability: displayed are of dose–response curves averaged over all 44 cell lines ($n = 88$ (duplicates of 44 cell lines), mean ± s.e.m.; colour code as in **c**). Hill slopes are indicated as dashed black lines. (**e**) Hill slope values of individual cell line viability curves (drugs added coloured as in **c**; ****depict $P < 0.0001$ (Friedmann's test with Dunn's multiple comparison) for comparison with PQR309, GDC0980, GDC0941. Indicated values are mean ± s.d. (**f**) Cross-correlation of cell lines' Hill slopes of indicated drug pairs. Dashed lines reflect the Hill slope ratio of 1 (for more results see Supplementary Fig. 1o). (**g**) Cross-correlation penalty score calculations based on Hill Slopes (see Methods). (**h**) Heat map depicting drug-induced fold changes of phospho-Histone H3-positive cells 24 h after drug treatment (all compounds at 2 μM; colchicine 200 nM) as compared to DMSO. Cell lines are sorted according to sensitivity to PQR309 ($n = 6$ for K562 and LS174-T $n = 3$). An overview of chemical formulas of relevant compounds is provided in Supplementary Fig. 1p. NS, nonsignificant.

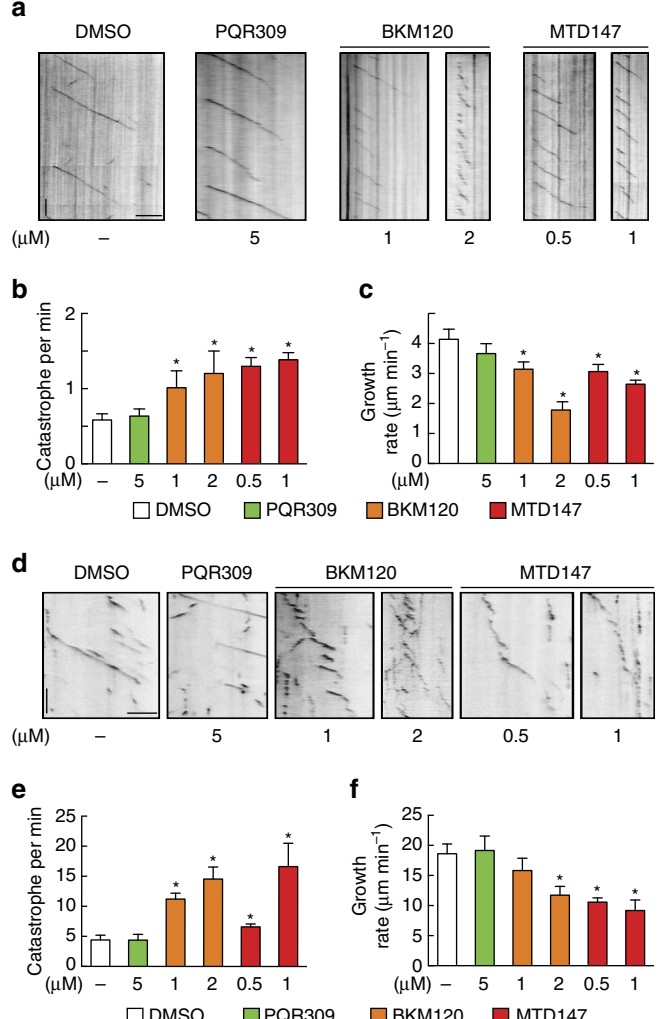

**Figure 2 | Drug-dependent changes in microtubule dynamics.**
(**a**) Representative kymographs depicting in vitro microtubule dynamics determined in the presence of the microtubule plus end tracking protein GFP-EB3 and the indicated drugs. To save space, kymographs were truncated horizontally to the longest microtubule traces. Time bar (vertical, 60 s) and distance bar (horizontal, 3 μm) apply to all images. (**b**) Quantification of the in vitro microtubule catastrophe frequency derived from experiments as exemplified in **a** in the absence or presence of indicated drugs (mean ± s.d., DMSO $n = 5$, compounds $n = 3$, $*P = 0.036$, Mann–Whitney test). (**c**) Quantification of microtubule growth rates in vitro (mean ± s.d., DMSO $n = 5$, compounds $n = 3$, $*P = 0.036$, Mann–Whitney test). (**d–f**) Microtubule dynamics were monitored in HeLa cells stably expressing plus-end binding EB3-GFP in the presence of PQR309, BKM120 and MTD147 using live cell microscopy and are illustrated as kymographs. Time bar (vertical, 20 s) and distance bar (horizontal, 3 μm) apply to all images. (**e**) Quantification of cellular microtubule catastrophe frequencies (mean ± s.d. DMSO $n = 5$, PQR309 $n = 4$, BKM120 and MTD147 $n = 3$; $*P = 0.0238$, Mann–Whitney test) in the presence of indicated drugs. (**f**) Quantification of drug effects on cellular microtubule growth rates (mean ± s.d. DMSO $n = 5$, PQR309 $n = 4$, BKM120 and MTD147 $n = 3$; $*P = 0.0238$, Mann–Whitney test).

stathmin-like protein RB3 (R) and tubulin tyrosine ligase (TTL; the complex is denoted $T_2R$-TTL[12,13]) were determined at resolutions of 2.05 and 2.25 Å, respectively (Supplementary Table 7), while PQR309 could not be soaked into $T_2R$-TTL crystals. The tubulin–ligand complex structures revealed that

both compounds bound in an indistinguishable fashion to the colchicine site[14] located between the α- and β-tubulin subunits (Fig. 3). It is well established that MDAs targeting this site inhibit the 'curved-to-straight' conformational change that must occur to allow free 'curved' tubulin to incorporate into 'straight' microtubules[14]. Our structural data (Fig. 3) thus classify BKM120 and MTD147 as colchicine-site MDAs and are consistent with cell cycle and microtubule dynamics data shown in Figs 1 and 2, as well as classical microtubule polymerization assays (Supplementary Fig. 2 and Supplementary Table 4).

Interactions between BKM120 and tubulin are mediated by an extensive network of direct and water-mediated contacts between polar and hydrophobic atoms (Fig. 3b). The core pyrimidine ring nitrogens are, however, devoid of any hydrogen bonding contacts. Despite its structural similarity to BKM120 and MTD147, PQR309 does not bind to tubulin. This suggests that the additional nitrogen in the core triazine of PQR309 is crucial to prevent its interaction with tubulin. At a resolution around 2 Å, a nitrogen cannot be distinguished from a C–H group in an aromatic ring and thus generates a pseudo-symmetry in the BKM120 molecule (Supplementary Fig. 3). This pseudo-symme-try obscures the definitive positioning of the core C–H group towards β-tubulin residues βLeu248 or βMet259 and thus prevents the full annotation of interactions.

To resolve this ambiguity, we produced asymmetric BKM120 derivatives by replacing each morpholino group with pyrrolidine to generate the corresponding regioisomers MTD265 and MTD265-R1 (Fig. 4a and Supplementary Fig. 4a,b). Remarkably, MTD265 induced mitotic arrest at 30 times lower concentrations than MTD265-R1 (Fig. 4b). The difference in potency of MTD265 versus MTD265-R1 as microtubule polymerization inhibitors could also be confirmed in vitro (Supplementary Fig. 4c–e and Supplementary Table 4).

Crystal structures of tubulin-MTD265 and tubulin-MTD265-R1 complexes at resolutions of 2.15 and 2.25 Å, respectively (Supplementary Table 7), clearly resolved that the morpholino moieties of MTD265 and MTD265-R1 point towards the guanine nucleotide-binding site, as in the case of BKM120 and MTD147 (Fig. 4c,d and Supplementary Fig. 4a,b). This result, in combination with the distinct biological activity of the two MTD265 regioisomers, defines the high-affinity orientation of the core pyrimidine with its C–H group pointing towards βMet259 (C–H in position V, see Fig. 4a). The higher potency of MTD265 as compared with MTD265-R1 can be explained by multiple differential contacts: (i) the formation of a favourable S···H–C aryl interaction[15] of MTD265's pyrimidine C–H group with the lone pair electrons of the βMet259 sulfur atom and (ii) hydrophobic interactions with the side chain of βAla316 (Fig. 4c). In MTD265-R1, the core nitrogen in position V points towards βMet259 and is expected to cause lone pair repulsions with the βMet259 sulfur atom and the backbone amide of βAla316.

Finally, the side chain of βLys352 is positioned such that it permits for a cation interaction between its ζ-nitrogen atom and the core π-system[16] of BKM120, MTD147 and MTD265 (Fig. 4d). The relocation of a core nitrogen to position V in MTD265-R1 (Fig. 4a,d) is likely to move electronegativity in the π-system away from the morpholino group, which might explain why the βLys352 is poorly resolved in tubulin-MTD265-R1 complex structures.

Taken together, these data suggest that high affinity binding of BKM120 to tubulin occurs via the core C–H group oriented towards βMet259. This conclusion readily explains why PQR309 (Fig. 2a–c and Supplementary Fig. 2a,b) and the BKM120 regioisomer (BKM120-R1; Supplementary Fig. 4g–j) do not perturb microtubule dynamics, as there is no possible orientation

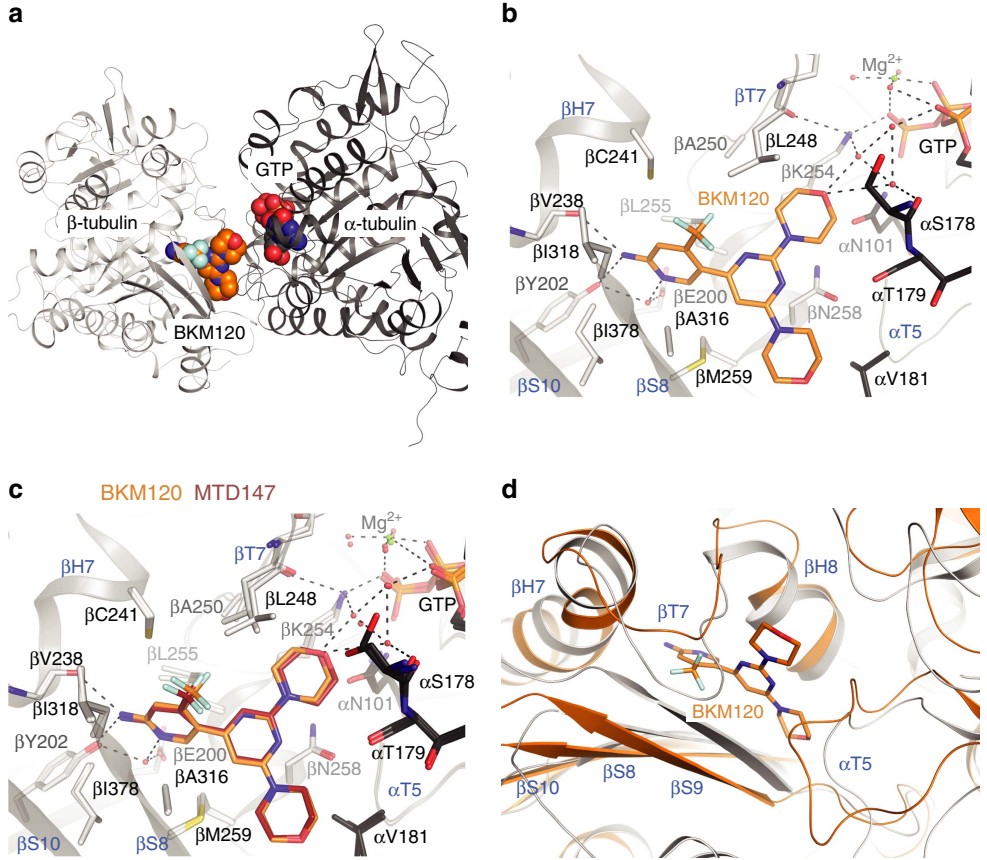

**Figure 3 | Interactions of BKM120 and derivatives with tubulin.** (**a**) Backbone of the αβ-tubulin heterodimer (in cartoon representation) in complex with BKM120; GTP is shown in spheres representation (PDB ID 5M7E). (**b**) BKM120 binding site in tubulin. Relevant amino acids are labelled in single letter code; secondary structure elements (marine blue) are H: helix; S: β-sheet; T: T-loop, preceded by the respective tubulin subunit, α or β. Resolved water molecules are indicated as red spheres; dashed lines denote hydrogen bonds or interactions discussed in the main text. (**c**) Overlay of BKM120 and MTD147 (PDB ID 5M7G) binding to the colchicine-binding pocket of tubulin in the $T_2R$-TTL complex. None of the compounds affected the global conformation of tubulin in the complex (rmsd 0.290 Å; 1941 Cα atoms) compared to the non-ligated $T_2R$-TTL complex (PDB ID: 4I55, refs 12,13). Residues of strands βS8 and βS9, loop βT7 and helices βH7 and βH8 of β-tubulin and of loop αT5 of α-tubulin form the boundaries of the binding site. For all investigated compounds, the trifluoromethyl substituted α-aminopyridine moiety pointed into the hydrophobic pocket outlined by side chains of βCys241, βLeu248, βAla250, βAla316, βIle318 and βAla354, with its amino group in H-bond distance to the βTyr202 OH and the βVal238 backbone carbonyl. The ring nitrogen is in H-bond contact to βGlu200 and βTyr202 through a water molecule. One morpholino group points towards the nucleotide-binding site, with the ether oxygen connected to two water molecules that establish an H-bond network to the side chains of Asn101 of α-tubulin, to Lys254 of β-tubulin, to the alpha and gamma-phosphates of the nucleotide and to the backbone carbonyl of Ser178 of α-tubulin. (**d**) Overlay of the BKM120 binding region in αβ-tubulin in the context of a microtubule ('straight' tubulin conformation, grey; PDB ID: 1JFF[35]) and in αβ-tubulin bound to BKM120 (orange).

to prevent repulsive forces between their core nitrogens and βMet259.

**Orientation of BKM120 core nitrogens define PI3K interaction.** The importance of the BKM120 pyrimidine core orientation for tubulin binding prompted us to revisit available PI3K-inhibitor complex structures and to analyse them for structural ambiguities. A 3.2 Å resolution crystal structure of the PI3Kγ catalytic subunit (p110γ) in complex with BKM120 has been previously reported[4]. At this resolution a precise positioning of the BKM120 molecule in the p110γ-binding site is not possible.

To validate the rotational orientation of BKM120 in PI3K, the asymmetrically substituted BKM120 derivatives PIKiN1 and PIKiN2, as well as their relevant regioisomers PIKiN1-R1 and PIKiN2-R1, were produced (Fig. 5a,c). To abrogate one of the two possible hydrogen bonds of BKM120 with the backbone amide of Val882 (default numbering for p110γ; Val851 in p110α), one morpholino group was substituted by a piperidine, where the morpholino oxygen was replaced by a –CH₂– group. PIKiN1 and

PIKiN1-R1 represent thus the simplest BKM120 derivatives that produce a functional asymmetry (Fig. 5a). Interestingly, PIKiN1 and PIKiN2 bound to recombinant p110γ, p110α, p110β and p110δ with a 7–30-fold higher affinity as compared to their regioisomers (Fig. 5b,d and Supplementary Table 5), which was also confirmed in a p110α activity assay measuring directly PtdIns(3,4,5)$P_3$ (Supplementary Fig. 6 and Supplementary Table 6). This difference in activity between regioisomers was also reflected in cellular assays specifically monitoring pan-PI3K, PI3Kγ and PI3Kδ inhibition (6–10-fold; Supplementary Fig. 5a–d). Assuming that the piperidine cannot interact with the backbone amide of Val882 and is pointed towards the solvent, this result demonstrates that the orientation of the pyrimidine core has a significant impact on PI3K inhibitor activity.

To clearly determine the orientation of the substituted morpholino group, a bulky chloromethyl-azetidin-methanol substitution was introduced: here the orientation of the pyrimidine core of PIKiN2 bound to p110γ could be unambiguously determined in a crystal structure resolved to 2.51 Å. Its core

 

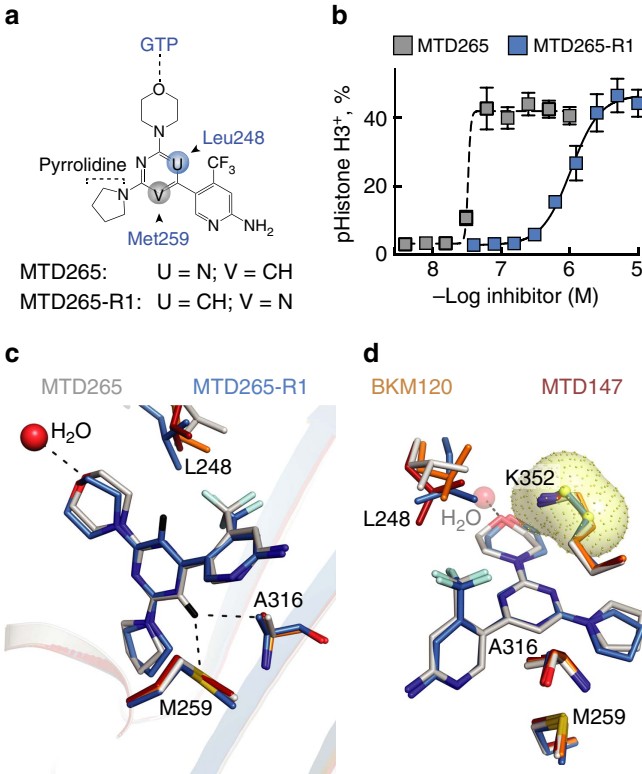

**Figure 4 | Determination of core pyrimidine orientation in tubulin complexes.** (**a**) Chemical formulas and schematic orientation of BKM120 regioisomeric derivatives MTD265 (PDB ID 5M8G) and MTD265-R1 (PDB ID 5M8D) in tubulin. (**b**) Phospho-Histone H3-positive A2058 cells triggered by increasing concentrations of MTD265 (grey) or MTD265-R1 (blue; % of total cells, $n = 3$, mean ± s.e.m.). (**c,d**) Structural overlay of amino acid side chains relevant for pyrimidine core ring interactions with BKM120 (orange), MTD147 (red), MTD265 (grey) and MTD-265-R1 (sky blue) in stick representation. The indicated water molecule is oriented towards the GTP-binding site (**c**) Visualization of βMet259 and βAla316 interactions with the pyrimidine core of MTD265 and MTD-265-R1 (core C–H protons shown in black). (**d**) βLys352 cation in proximity to pyrimidine core π-system of MTD265 and MTD-265-R1. The βLys352 side chain is fully resolved in crystal structures of BKM120, MTD265 and MTD147, but is poorly defined in MTD265-R1 complex beyond the δCH2 of βLys352 (yellow spheres and cloud denote poorly defined atoms).

pyrimidine C–H group points towards Tyr867 (position U in Fig. 5e and Supplementary Table 8). This result clearly opposes the crystal structure of the p110γ–BKM120 complex previously reported by Burger *et al.*[4] and models of BKM120–p110α complexes derived thereof[3]; all these studies positioned a core nitrogen towards the tyrosine Tyr867 in p110γ.

To test whether the C–H core group oriented towards Tyr867 is required for PI3K inhibition, we produced the BKM120 regioisomer (BKM120-R1; Supplementary Fig. 4g) and PI3KiN3 (Fig. 5c), which always point a core nitrogen towards Tyr867 of p110γ. Both modifications had little effect on PI3K inhibitor affinities (Supplementary Table 5, Fig. 5d and Supplementary Fig. 5e–g), suggesting that the core nitrogens, but not the C–H group, are required for the interaction with PI3K. This conclusion is further supported by a 2.48 Å resolution p110γ–PI3KiN3 complex structure, which displays an identical binding mode for PI3KiN3 as for PI3KiN2 (Fig. 5f and Supplementary Table 8).

Closer inspection of the p110γ–PI3KiN2 and p110γ–PI3KiN3 complex structures revealed that (i) the core nitrogen in position V is hydrogen bonded through a structured water molecule to Asn951 and Asp964, and (ii) that the trifluoromethyl-pyridin-2-amine group and aspartates 841, 836, 964 coordinate two additional structured water molecules (best resolved in the PI3KiN3 complex; Fig. 5e,f). Molecular dynamics simulations with BKM120 docked into p110γ demonstrated that positioning of the pyrimidine core C–H group towards Tyr867 kept the structured water network intact ($C_U$–H; Fig. 5g). In contrast, when BKM120 was oriented as in the initially reported structure by Burger and colleagues[4] ($C_V$–H; Fig. 5h), hydrogen bond interactions between the inhibitor, water molecules, side chains of Asp964 and Asp836 were disrupted. These data suggest that high-affinity PI3K binding of BKM120 and its derivatives only occurs when a core nitrogen (in the V position) interacts with a structured water molecule, which readily explains why MTD147 without a suitable core nitrogen loses potency as a PI3K inhibitor.

**Clinical BKM120 levels—relation to biological action.** The separation of BKM120's activities into the PI3K inhibitor PQR309 and the MDA MTD147 allows a deeper analysis of the drug's effects at therapeutically relevant concentrations. With PQR309, PI3K inhibition of 87–95% is required to inhibit proliferation by 50% in most cell lines. For MTD147, the same effect is achieved with as little as 13–28% of the maximal achievable Histone H3 phosphorylation (Fig. 6a,b), which is in agreement with findings that cytotoxicity is already caused when 2–20% of the total tubulin pool is bound to microtubule-binding compounds[17]. BKM120 yielded 87–95% PI3K inhibition at $1.4 \pm 0.8 \mu M$, while the required concentration to reach 13–28% of MDA activity was $0.8 \pm 0.2 \mu M$. Both values are close to the observed $IC_{50}$ for proliferation reached at $0.9 \pm 0.2 \mu M$ (Supplementary Fig. 1d), thus leaving no window for selective PI3K inhibition. Similarly, a concentration of $1.2 \mu M$ BKM120 is required for median half-maximal inhibition of proliferation of 44 different cell lines (see Fig. 1 and Supplementary Table 2). These values match previously published data for 282 cells lines or primary cells. Seventy-five per cent of all cell lines in each panel have an $IC_{50} \geq 0.9$ and $0.75 \mu M$, respectively (Fig. 6d). This illustrates that most cell lines only show significantly attenuated proliferation at BKM120 concentrations where the drug affects microtubule dynamics.

In patients, mean BKM120 plasma concentrations range from 1.5 to $2 \mu M$ at 50 mg per daily dosing (q.d.) and $2–3 \mu M$ at 100 mg q.d. as calculated from published values (area under the curve, $AUC_{0–24h}$; refs 6,18). These drug levels strongly suggest that BKM120 also targets microtubules dynamics during therapy (Fig. 6c) and that on/off-target activities cannot be selectively controlled.

**Discussion**
BKM120 has excellent pharmacological properties, is highly selective for the class I PI3K family versus protein kinases and is one of the few PI3K inhibitors that readily crosses the blood brain barrier[3–6]. This provides additional therapeutic opportunities to target malignant cells in the brain, but exposes this organ to BKM120 off-target actions. Dose-limiting toxicities caused by MDAs in the brain are difficult to evaluate in animal models[7], a fact that emphasizes the importance of an exact understanding of molecular mechanisms of drug actions.

We succeeded in separating the activities of BKM120 as a PI3K inhibitor and an MDA by minimal chemical modification of its core pyrimidine ring. The resulting selective PI3K inhibitor PQR309 and the MDA MTD147 thus allowed a profiling of BKM120 against the closest possible reference compounds.

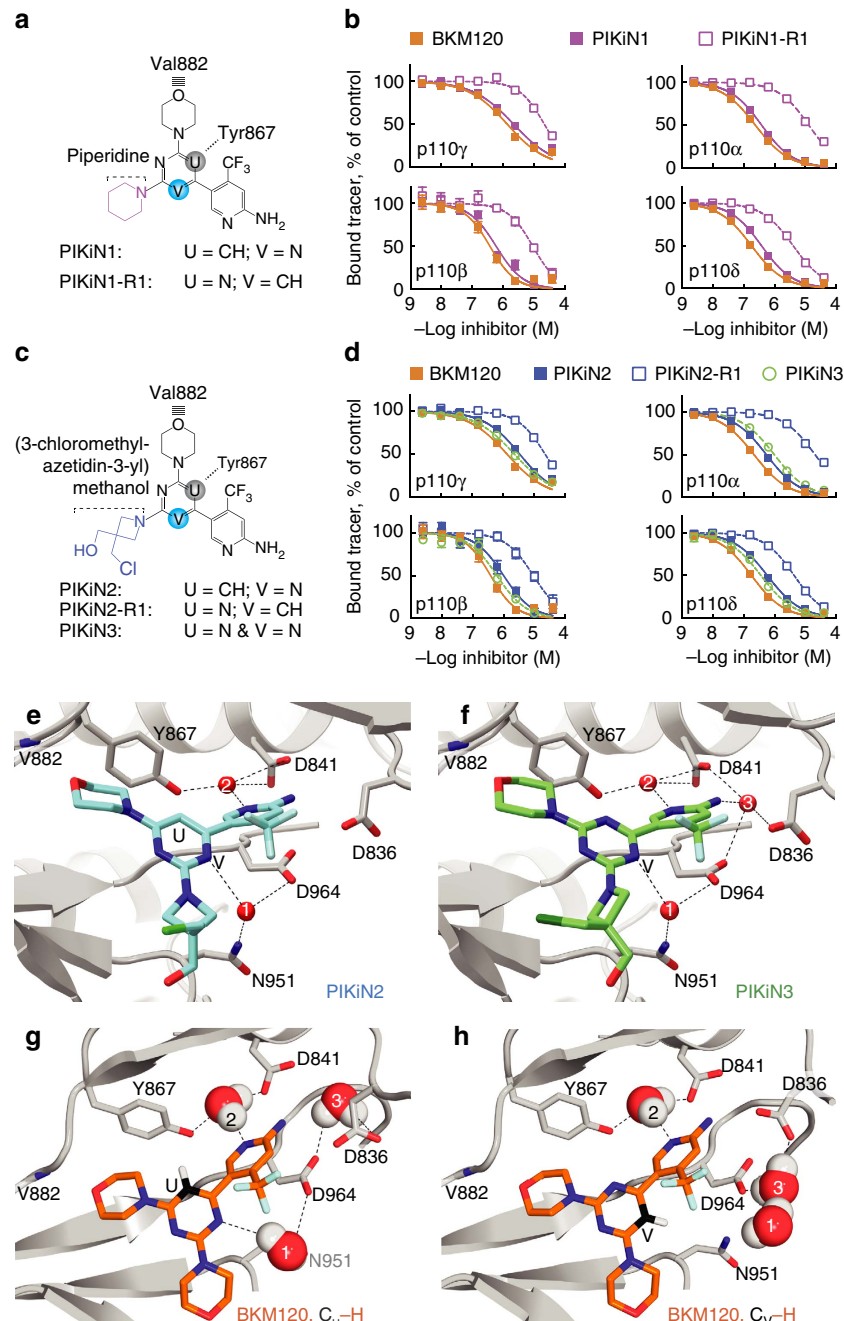

**Figure 5 | Binding of asymmetric BKM120 derivatives to PI3Ks.** (**a,c**) Chemical formulas of regioisomeric pairs. Val882 and Tyr867 of p110γ indicate schematically the drugs orientations. (**b,d**) PI3K isoform-specific competitive binding assays used for $K_d$ calculations in Supplementary Table 4 ($n \geq 2 \times 2$ (details in Methods); error bars omitted when smaller than symbols). (**e,f**) Co-crystal structure of p110γ soaked with PIKiN2 (**e**, (PDB ID 5JHA)) and PIKiN3 (**f**, (PDB ID 5JHB)), depicting structured water molecules (red numbered spheres), and water coordinating amino acids with hydrogen bonds as dashed lines. (**g,h**) Two opposing orientations of BKM120 in p110γ were set as starting points for modelling of water movements. Positions of water dipoles after molecular dynamics calculations and energy minimization are shown. The BKM120 structure (from PDB ID 3SD5) was fitted into the protein/water scaffold of the PIKiN3–p110γ complex before molecular dynamics calculations. Water molecules, but not BKM120 and protein, were allowed to move during calculations.

The analysis of cancer cell line sensitivity and cell cycle profiles, hill slopes of dose–response curves, PI3K and MDA activities all revealed that the cellular activity of BKM120 matches the mode of action of MTD147 and known microtubule disruptors such as nocodazole and colchicine. This similarity of BKM120 to other MDAs connects its dominant cellular action with a mitotic arrest by directly interacting with tubulin dimers to perturb microtubule dynamics.

Our structural analysis of the BKM120–tubulin interaction elucidates how the orientation of the BKM120 molecule within its binding site is governed by hydrophobic interactions, and illustrates that a rotation of BKM120 by 180° switches between a high- and a low-affinity conformation as corroborated by the structures and activities of MTD265 and MTD265-R1. Interestingly, the same is true for PI3K: here a water network stabilizes inhibitor contacts with evolutionarily conserved amino acid side

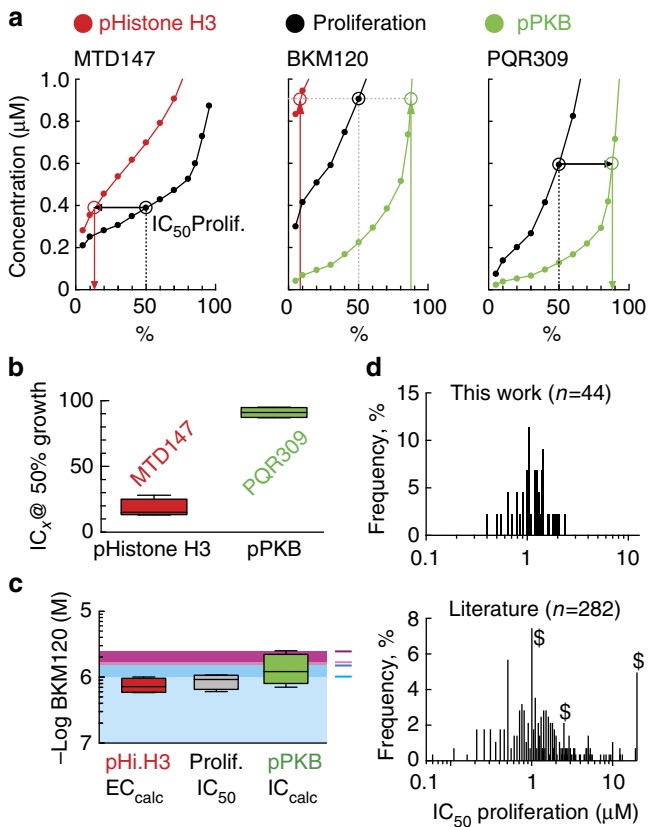

**Figure 6 | BKM120 and derivative drug actions at therapeutic doses.**
(**a**) Plot of MTD147, BKM120 and PQR309 concentrations versus Histone
H3 phosphorylation (% of maximal; red curves), proliferation (black) and
PKB/Akt phosphorylation (pPKB, % of total; green). pHistone H3
phosphorylation and pPKB/Akt levels matching 50% growth inhibition are
indicated by arrows. (**b**) Levels of phospho-Histone H3 for MTD147 and
% of inhibition of pPKB/Akt for PQR309 correlating with 50% growth
inhibition are shown as mean of A2058, SKOV3, BT549 and U87-MG cells
responses. (**c**) Calculated concentrations of BKM120 required to achieve
50% growth when associated with either phospho-Histone H3 (red) or
PI3K inhibition (green). The determined $IC_{50}$ for proliferation for BKM120 is
shown in grey. Shaded areas indicate patient BKM120 plasma levels
reported in ref. 6. Colour code: blue is 50 mg per daily (light: after day 8 of
cycle 1; dark: after day 1 in cycle 2); magenta is 100 mg per daily (light: after
day 8 of cycle 1; dark: after day 1 in cycle 2). Box and whiskers; whiskers min
to max. (**d**) Frequency distribution of BKM120's half maximal concentration
for proliferation per cell viability in the 44 cell lines (*top*: first bin centre
0.1 μM, last bin centre 12.1 μM, bin width 0.05 μM). *Bottom*: Meta-analysis
of reported BKM120's $IC_{50}$ concentrations (μM) for proliferation/cell
viability of 282 cell lines or primary cells, illustrated as frequency
distribution (first bin centre 0.01 μM, last bin centre 20.01 μM, bin width
0.05 μM). $ indicates bins containing values that were reported in literature
as $IC_{50} > x$ μM ($x =$ position of indicated bin). Values were extracted from
refs 36–47.

chains, which include the ATP-binding DFG motif. Structured
water is present in other PI3K and protein kinase structures, but
was rarely considered in rational drug design. Mutations affecting
water networks might, however, contribute to loss of drug action,
as it has been recently observed for the BCR-Abl inhibitor
bosutinib: in this case, water-mediated hydrogen bonds at the
DFG motif of the kinase were reported to determine drug
selectivity and resistance[19].

The precise structural and chemical dissection of the drug's
dual activity allowed a reassessment of the respective target

occupancy required to inhibit cell proliferation: the availability
of MTD147 and PQR309 shows that cellular phenotypes
associated with disruption of microtubule dynamics require low
EC values, while PI3K needs to be inhibited >90% to achieve
a 50% inhibition of cell proliferation. In this context, our results
strongly suggest that BKM120 blocks cell proliferation mainly via
microtubule disruption and not through PI3K inhibition.

BKM120 plasma concentrations in patients undergoing
continuous dosing have been reported by Bendell *et al.*[18] and
Saura *et al.*[6] and exceed levels required to trigger MDA activity.
One might therefore expect that BKM120 concentrations at
steady state in repetitive daily dosing protocols effect microtubule
dynamics, and that these contribute to antitumour activity and
adverse effects of the drug.

It remains presently elusive if the mixed PI3K/MDA profile of
BKM120 is preferable over a pure PI3K inhibitor in clinical
settings. Indeed, a variety of ongoing clinical trials already test
combinations of PI3K and microtubule inhibitors for efficacy in
cancer treatment (clinicaltrials.gov). Current treatment schemes
involve dosing of microtubule-targeting drugs such as Paclitaxel
at 1–3 week intervals, while PI3K inhibitors are applied on a daily
basis. Our studies have shown how the bipartite activities of
BKM120 can be separated and can provide flexible dosing
schemes: the process has produced PQR309, a specific PI3K
inhibitor devoid of MDA activity, which is now in phase II
clinical trials. The highly potent MDA MTD265 demonstrates
that BKM120 can serve as a seed molecule for novel colchicine-
site binding MDAs.

In clinical settings where combined targeting of PI3K and
microtubule dynamics proves to be beneficial, distinct PI3K
inhibitors and MDAs provide flexibility in drug scheduling
schemes. Furthermore, in such a setting specific drug-associated
toxicities and adverse effects can be managed individually
for each target. If united in one drug, dose adaptations of
BKM120 would keep PI3K/MDA activity ratios on a linear
trajectory without the possibility to control target saturation
individually. This means that once a maximally tolerated dose is
reached for one target, a full therapeutic access to the second
mode of action is limited by the target with the lower maximally
tolerated dose (Fig. 7).

PI3K inhibitors and MDAs target similar processes associated
with tumour promotion, such as tumour cell growth and
proliferation, but they also affect endothelial function and
angiogenesis. As the molecular mechanisms to achieve these
therapeutic effects are unrelated, beneficial synergisms and
suppression of drug resistance are the motivation of current
combinatorial drug trials.

In a randomized double-blind study (BELLE-4) BKM120/
buparlisib has been combined with paclitaxel in human epidermal
growth factor receptor 2-negative (HER2) breast cancer patients.
The interim analysis of progression-free survival (PFS) data
of paclitaxel/buparlisib versus paclitaxel/placebo in this phase
II/III study did not warrant the extension into phase III[20].
Buparlisib has previously shown encouraging clinical activity
as a single agent[21]. In the BELLE-4 study, the paclitaxel/buparlisib
combination also failed to show an increase in progression-free
survival in patients with an activated PI3K pathway. Although
it is tempting to speculate that this outcome is linked to the
action of buparlisib as an MDA, dosing schedules, pharmaco-
logical parameters, patient selection, etc. could dominate the
negative clinical outcome. This seems currently to be in
agreement with the phase II PEGGY-study in HER2⁻ patients,
where paclitaxel was combined with the non-brain penetrable
pan-PI3K inhibitor pictilisib (GDC-0941) or placebo, and also did
not reveal a significant benefit of the combinatorial treatment[22].
The active molecules in the BELLE-4 and PEGGY-study are,

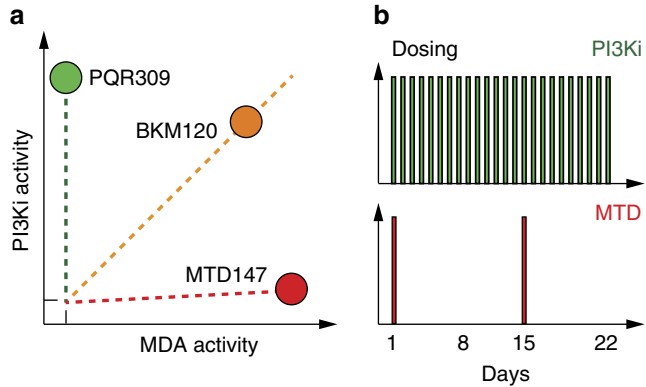

**Figure 7 | Schematic activity map of BKM120 and BKM120 derivatives and dosing schemes.** (**a**) BKM120 targets PI3K and tubulin proportionally. A concentration increase of BKM120 does therefore not change PI3K/tubulin targeting ratios, while combinations of selective PI3K inhibitors (such as PQR309) and potent microtubule-targeting drugs (MTDs) allow a flexible access to PI3K inhibition and perturbation of microtubule dynamics. (**b**) PI3K inhibitors are typically administered daily (dosing schedule in green), while microtubule targeting drugs are usually given in 1–3 week intervals (dosing schedule in red) for a limited time only.

however, structurally unrelated and have different physico-chemical properties. While there are a number of multi-targeted drugs on the market, advanced clinical studies with structurally closely related, functionally redefined derivatives are not available. It will be interesting to follow the progression of BKM120 and PQR309 through clinical development analysing differential responses and adverse effects.

## Methods

**Cell culture.** A2058, SKOV3, BT-549, U87-MG and HCT-116 cells (originating from ATCC) were grown in complete Dulbecco's modified Eagle's medium (DMEM) (DMEM supplemented with 10% heat-inactivated fetal calf serum, 2 mM L-glutamine and 1% penicillin–streptomycin; Sigma) at 37 °C, 5% $CO_2$. Cell lines were tested negative for mycoplasma. Cell line identity was confirmed using highly polymorphic short tandem repeat loci (STRs) profiling (Microsynth, Switzerland). Bone marrow-derived mast cells (BMMCs) were isolated, differentiated and grown slightly modified from ref. 23. In short, cells from fresh murine bone marrow were suspended in complete Iscove's modified Dulbecco's medium (IMDM) with 10% heat-inactivated fetal calf serum, 2 mM, L-glutamine, 1% penicillin–streptomycin solution, 50 μM β-mercaptoethanol and recombinant murine IL-3 (2 ng ml$^{-1}$; Peprotech) and recombinant murine stem cell factor (5 ng ml$^{-1}$; Peprotech), and cultured at 37 °C and 5% $CO_2$ for 4 days. Subsequently, BMMCs were diluted weekly to $0.5 \times 10^6$ cells per ml with a mixture of 80% fresh, complete IMDM and 20% recycled medium, with IL-3 added every 2–3 days. Differentiation was monitored by detection of FcεRI (with a phycoerythrin-conjugated hamster antibody to mouse FcεRIα; clone MAR-1; eBioscience #12-5898-83) and c-kit (with a Allophycocyanin-conjugated anti-mouse CD117/c-kit antibody; clone 2B8; Biolegend #105812) by fluorescence-activated cell sorting analysis.

**In-cell western detection of phosphorylated PKB/Akt.** $2 \times 10^4$ cells per well were seeded in 96-well plates (Cell Carrier; Perkin Elmer) the day before inhibitor treatment. Inhibitors (or dimethyl sulfoxide (DMSO)) were added for 1 h (37 °C, 5% $CO_2$) at indicated concentrations. Subsequently cells were fixed in paraformaldehyde (PFA; 4%) in phosphate-buffered saline (PBS) for 30 min at room temperature (RT), and then blocked with 1% bovine serum albumin (BSA)/0.1% Triton X-100/5% goat serum in PBS (30 min, RT). Primary antibodies (rabbit anti-phospho-Ser473 of PKB/Akt from Cell Signaling Technology # 4058; mouse anti-α-tubulin, Sigma # T9026; diluted 1:500 and 1:2,000, respectively) were added overnight (4 °C, shaking). The next day, three wash cycles of 5 min with 1% BSA/0.1% Triton X-100 in PBS were followed by incubation with IRDye680-conjugated goat anti-mouse, and IRDye800-conjugated goat anti-rabbit antibodies for 1 h (LICOR # 926-68070 and # 926-32211, shaking, RT, in the dark), and later washed 5 × 5 min with 1% BSA/0.1% Triton X-100 in PBS. Fluorescence was measured with an Odyssey infrared imaging scanner (LICOR; on 'In Cell Western' mode, offset 4.0 μm, automatic exposure for both channels, using Image Studio

Ver4.0 software from LICOR). Cells exposed to DMSO represent 100% controls, and wells without primary antibody are defined as signal background. Percentage of remaining phospho-Ser473 PKB signal was calculated including a correction for cellular protein, as determined by tubulin staining. Values were displayed as a function of inhibitor concentration (log scale) using GraphPad Prism. $IC_{50}$ values of inhibitors were determined by fitting with GraphPad's normalized nonlinear regression curve function with variable slope:

$$y = 100 / \left(1 + 10^{[(\text{Log IC}_{50} - \text{Log} x) \times \text{HillSlope}]}\right), \text{ where } x = \text{drug concentration}$$

Data were assessed for A2058, BT-549 in 5, for HCT116 in 4, and for SKOV3 and U87-MG cell lines in three independent experiments.

**High-content/high-throughput microscopy.** *Cell preparation.* A2058, SKOV3, BT-549, U87-MG and HCT-116 cells were seeded in 100 μl complete DMEM into 96-well plates (Cell Carrier; Perkin Elmer) the day before inhibitor treatment. Cells were exposed to inhibitors (ranging from 10–0.01 μM, for colchicine and nocodazole: 1–0.001 μM) for indicated times, and fixed by addition of half a volume 10% PFA in PBS (30 min, RT). Cells were permeabilized and stained with Hoechst33324 by addition of 1:10 volume of 1% BSA/1% Triton X-100, 22 μM Hoechst33324 in PBS (30 min, RT), followed by a first image acquisition, see below. Subsequently cells were stained overnight (4 °C, on a shaker) with rabbit anti-phospho-serine 10 Histone H3 (Cell Signaling Technologies, # 9701) and rat anti-α-tubulin antibodies (Clone YL1/2; Santa Cruz Technologies, # sc-53029). Cells were then washed and incubated with Alexa488-conjugated goat anti-rat IgG, Alexa647-conjugated goat anti-rabbit IgG antibodies (Life Technologies # A-11006 and # A-21245) and 2.2 μM Hoechst33324 in 1% BSA/0.1% Triton X-100/PBS (shaking; RT; dark), followed by three wash cycles and a second round of imaging.

*Image acquisition.* Fluorescent microscopy images were acquired on an Operetta high content analysis system (Perkin Elmer). Thirty-five fields of view/well were acquired either in confocal mode (z-stacks, × 20 high NA objective, applied in early experiments) or alternatively as wide field images (20x WD objective). 1st *acquisition:* Nuclear DNA content and DNA morphology were imaged using the DAPI filter set. 2nd *acquisition:* Nuclear DNA, phosphorylated Histone H3 and α-tubulin images were acquired using DAPI, Alexa488 and Alexa647 filter sets.

*Image analysis.* Images were batch analysed using Columbus software (PerkinElmer). In case of confocal acquisitions, z-stacks were combined by maximum projection. Analyses were run including single-cell analysis mode. The total number of nuclei was determined in the DAPI channel. Nuclear DNA was classified into 'normal' or 'condensed' using a linear classifier mode. The percentage of cells with condensed DNA was calculated for each well with

$$\% \text{ condensed} = (\# \text{ cells with condensed DNA} / \# \text{ total cells}) \times 100),$$
$$\text{where } \# \text{ is 'number of'}$$

Hoechst33324 intensity of each nucleus was further used to calculate cell cycle distribution in each well using R script.

The fraction of cells with phospho-Histone H3-positive nuclei was determined using Hoechst33342 and Alexa647 signals. Nuclei were first stained for DNA as described above (Hoechst33342), and nuclei at image borders were excluded from calculations. The remaining nuclei were classified as positive or negative for phospho-Histone H3 by a two-step selection using the 'filter by properties' mode based on Alexa647 intensities. Alexa647 filter properties were (i) mean intensities above threshold/nucleus and (ii) a maximum intensity above threshold. Calculation was performed with the formula:

$$\% \text{ Alexa647}^{\text{positive}} = (\# \text{ Alexa647}^{\text{positive}} / \# \text{ selected nuclei}) \times 100,$$
$$\text{where } \# \text{ is 'number of'}$$

The percentage of cells with condensed DNA, the percentage of phospho-Histone H3-positive nuclei and the total number of nuclei (normalized to DMSO control as a measure for proliferation inhibition at 72 h) were plotted as a function of inhibitor concentration (log scale) using GraphPad Prism. $EC_{50}/IC_{50}$ values of inhibitors were determined using GraphPad's 'non-linear regression with variable slope' formula:

$$y = \text{bottom} + (\text{top} - \text{bottom}) / \left(1 + 10^{[(\text{Log EC}_{50} - \text{Log} x) \times \text{HillSlope}]}\right)$$

**Ratio of mitotic arrest and PI3K inhibition.** Concentrations of half-maximal pSer473 PKB inhibition was determined as described under 'In Cell Western'. Half-maximal effective concentrations for appearance of phospho-Histone H3-positive cells were determined as described under 'high-throughput microscopy'. Five cell lines (A2058, BT549, SKOV3, U87-MG and HCT116) were subjected to both analyses in at least three independent experiments. Subsequently, the ratios of the mean $EC_{50}$ for phospho-Histone H3 to the mean $IC_{50}$ for pSer473 PKB for each cell line and drug (PQR309, BKM120, MTD147) were calculated. For PQR309 the $EC_{50}$ pHistone H3 was set constant to 20 μM, since no increase in mitotic cell population was detectable with PQR309.

**Viability studies on 44 cell lines.** *Compound preparation.* Compounds were weighed on a calibrated balance and dissolved in 100% DMSO. DMSO samples were stored at room temperature. At the day of the experiment, the compound stock was diluted in 3.16-fold steps in 100% DMSO to obtain a 9-point dilution series. This was further diluted 31.6 times in 20 mM sterile Hepes buffer pH 7.4. A volume of 5 µl was transferred to the cells to generate a test concentration range from $3.16 \times 10^{-5}$ to $3.16 \times 10^{-9}$ M in duplicate. The final DMSO concentration during incubation was 0.4% in all wells.

*Cell proliferation assay.* The 44 cell lines have been authenticated at the American Type Culture Collection (ATCC) and have been licensed by NTRC (Netherlands). Of the 44 cell lines in this commercial cell panel only BT-20 and J82 are listed by the International Cell Line Authentication Committee as eventually misidentified. Cell lines were used for experiments below nine cell passages. Assay stocks of 44 cell lines were thawed and diluted in its ATCC recommended medium and dispensed in a 384-well plate, depending on the cell line used, at a concentration of 400–1,600 cells per well in 45 µl medium. The margins of the plate were filled with PBS. Plated cells were incubated in a humidified atmosphere of 5% $CO_2$ at 37 °C. After 24 h, 5 µl of compound dilution was added and plates were further incubated for another 72 h. After 72 h, 25 µl of ATPlite 1Step (PerkinElmer) solution was added to each well, and subsequently shaken for 2 min. After 10 min of incubation in the dark, the luminescence was recorded on an Envision multimode reader (PerkinElmer).

*Controls.* $T = 0$ signal. On a parallel plate, 45 µl cells were dispensed and incubated in a humidified atmosphere of 5% $CO_2$ at 37 °C. After 24 h 5 µl DMSO-containing Hepes buffer and 25 µl ATPlite 1Step solution were mixed, and luminescence measured after 10 min incubation ($= \text{luminescence}_{t=0}$).

*Reference compound.* The $IC_{50}$ of the reference compound doxorubicin is measured on a separate plate. The $IC_{50}$ is trended. If the $IC_{50}$ is out of specification (0.32–3.16 times deviating from historic average) the assay is invalidated.

*Cell growth control.* The cellular doubling times of all cell lines are calculated from the $t = 0$ h and $t = 72$ h growth signals of the untreated cells. If the doubling time is out of specification (0.5–2.0 times deviating from historic average) the assay is invalidated.

*Maximum signals.* For each cell line, the maximum luminescence was recorded after incubation for 72 h without compound in the presence of 0.4% DMSO ($= \text{luminescence}_{\text{untreated}, t=72h}$).

*Data analysis.* $IC_{50}$s were calculated by nonlinear regression using IDBS XLfit 5. The percentage growth after 72 h (%-growth) was normalized as follows: $100 \times (\text{luminescence}_{t=72h}/\text{luminescence}_{\text{untreated}, t=72h})$. This was fitted a 4-parameter logistics curve:

$$\text{\%-growth} = \text{bottom} + (\text{top} - \text{bottom})/(1 + 10^{[(\text{Log IC}_{50} - \text{Log } x) \times \text{HillSlope}]}),$$

where bottom and top are the asymptotic minimum and maximum cell growth that the compound allows in that assay.

**Hill slope determination and penalty score calculations.** *Hill slope steepness* was calculated for (i) drug-concentration-dependent cell viability curves measured at NTRC (44 cell lines, 72 h) and for (ii) drug-concentration-dependent cell numbers after 72 h using high-throughput microscopy (A2058, SKOV3, BT-549, U87-MG and HCT-116 cells). Hill slope steepness for each drug and cell line was determined by fitting a nonlinear regression curve with variable slope to the data using the GraphPad Prism function, with x as variable drug concentration:

$$y = \text{bottom} + (\text{top} - \text{bottom})/\left(1 + 10^{[(\text{Log IC}_{50} - \text{Log } x) \times \text{HillSlope}]}\right)$$

Cell lines (in i) or experiments in (ii) showing for at least one drug high uncertainty of slope steepness (defined by GraphPad Prism as 'ambiguous') were excluded from analysis.

Statistical analysis of the 36 remaining cell lines from NTRC was performed using Friedmann's test with Dunn's multiple comparison (because normal distribution was tested negative by D'Agostino-Pearson omnibus normality test for at least one compound). Statistics for the five cell lines with at least two independent experiments for cell number determination were calculated by one-way ANOVA with Tukey's multiple comparison tests (normal distribution tested by D'Agostino-Pearson omnibus normality test).

*Penalty scores:* penalty scores for cross-correlation of drug pairs are depicted as sums of individual penalty scores of all 36 included cell lines. For each individual cell line and drug pair penalty scores (ps) were calculated using

$$\text{ps} = 2 \times \left(\text{HillSlope}_{\text{drug1}} - \text{HillSlope}_{\text{drug2}}\right)^2$$

**Histone H3 phosphorylation in 44 cell lines.** Indicated cell lines were cultured in 384-well plates (CellCarrier Ultra) for 18 h in ATCC-recommended media, and then exposed to 2 µM PQR309, BKM120, MTD147, GDC0941, GDC0980 and 200 nM colchicine for 24 h (37 °C, 5% $CO_2$) at NTRC (Netherlands). Cells were fixed by addition of 10% PFA/PBS (final PFA concentration 3.3%). Plates were subsequently stained for phospho-Ser10 of Histone H3, α-tubulin and DNA content, and analysed as described under *high throughput microscopy* ($n = 6$; for

K562 and LS174-T $n = 3$). Cell lines growing loosely adherent or in suspension (CCRF-CEM, Jurkat-E6-1, MOLT4, SHP-77) were excluded from the analysis.

***In vitro* microtubule plus end tracking assay.** Reconstitution of plus end tracking *in vitro* was performed as described previously[24]. Short microtubule seeds were prepared by incubating the tubulin mix containing 70% unlabeled porcine brain tubulin (Cytoskeleton), 17% biotin-tubulin (Cytoskeleton), 6% rhodamine-tubulin (Cytoskeleton) with 1 mM GMPCPP (Jena Biosciences) at 37 °C for 30 min. Microtubules were depolymerized on ice for 30 min and repolymerized at 37 °C with 1 mM GMPCPP. The seeds were diluted 10-fold in assay buffer (80 mM Pipes, 4 mM $MgCl_2$, 1 mM EGTA, pH adjusted to 6.8 with KOH) containing 10% glycerol, snap frozen in liquid nitrogen and stored at $-80$ °C.

For the reconstitution assay, flow chambers were incubated first with 0.2 mg ml$^{-1}$ PLL-PEG-biotin (Susos AG, Switzerland), washed with the assay buffer and then incubated with 1 mg ml$^{-1}$ neutravidin. Immobilized seeds were attached to functionalized glass coverslips by biotin-neutravidin links. Flow chambers were further incubated with 1 mg ml$^{-1}$ κ-casein to prevent non-specific protein binding. The reaction mix of drugs, assay buffer supplemented with 15 µM porcine brain tubulin, 50 mM KCl, 1 mM GTP, 0.2 mg ml$^{-1}$ κ-casein, 0.1% methylcellulose and oxygen scavenger mix (50 mM glucose, 400 µg ml$^{-1}$ glucose oxidase, 200 µg ml$^{-1}$ catalase and 4 mM DTT) was added to the flow chamber. Experiments were carried out in the presence of 100 nM GFP-EB3. Time-lapse images of microtubule dynamics were acquired at 30 °C using TIRF microscopy.

Live cell imaging (37 °C) was performed on a Nikon Eclipse Ti-E microscope with the perfect focus system (Nikon), equipped with Nikon CFI Apo TIRF 100x 1.49 NA oil objective (Nikon), Photometrics Evolve 512 EMCCD (Roper Scientific) and CoolSNAP HQ2 CCD camera (Roper Scientific), and controlled with MetaMorph 7.7.5 software (Molecular Devices). A stage top incubator (INUBG2E-ZILCS; Tokai Hit) controlled temperature.

*Analysis of microtubule plus end dynamics in vitro.* Maximum intensity projections and kymographs were made using Metamorph. Microtubule dynamics parameters were determined from kymographs using the JAVA plug in for Image J described previously[24,25]. Each kymograph was split into alternating phases of growth and shortening. Growth of a microtubule starting from the seed until the catastrophe was considered as a single growth event, and the rate, time and length of each event were calculated. The length and the duration of each phase were measured as horizontal and vertical distances on the kymograph, respectively. The average velocity was calculated as a ratio of these values. Catastrophe frequency was calculated as the inverse growth time. The rescue frequency was calculated by dividing the number of observed rescues by the total shortening time. Only events with clearly observed start and end were taken into consideration during the analysis. Growth events shorter than 0.5 µm were not included in the analysis.

*Analysis of microtubule plus end dynamics in HeLa cells.* HeLa cells stably expressing EB3-GFP[26] were grown on coverslips for 16 h. Cells were treated with different drug concentrations and imaged at 0.5 s per frame in stream acquisition mode using the same TIRF microscope, 30 min after drug addition. The TIRF microscope was used in a semi-TIRF mode which allowed optimal visualization of the $\sim 0.5$–1-µm-thick part of the cell proximal to the coverslip. Microtubule dynamic parameters were measured from kymographs made using Metamorph. Dynamic parameters were measured only for MTs in the internal part of the cell and not at the cell cortex. To measure growth rates, the velocity of microtubule end displacements longer than 0.5 µm were taken into account. Catastrophe frequency per minute was calculated by dividing the total number of growth events by the total growth time in minutes.

*Statistical analysis.* The growth rate and catastrophe frequency in each condition was calculated by averaging over all microtubules in a movie for *in vitro* and by averaging over all microtubules per cell in case of HeLa cells. The differences between the microtubule dynamics parameters for control and conditions with drug treatment in the presence of GFP-EB3 *in vitro* and in HeLa cells stably expressing EB3-GFP were compared using a non-parametric Mann–Whitney test. Analysis was made using the GraphPad Prism version 6.04 for Windows (GraphPad software, La Jolla, CA, USA). *In vitro* tracking: DMSO $n = 5$, each drug $n = 3$; cellular microtubule dynamics: DMSO $n = 6$, PQR309 $n = 4$, BKM120 and MTD147 $n = 3$.

***In vitro* tubulin polymerization assay.** Cell-free tubulin polymerization assays were carried out with kit #BK006P from Cytoskeleton (Denver, USA) according to the manufacturer's instructions. Ice cold tubulin solution (100 µl, 27.5 µM in supplied polymerization buffer) was added to pre-heated (37 °C) 96-well plates containing 10 µl of inhibitors dissolved in the same buffer. Tubulin polymerization was then monitored by turbidity changes at 340 nm ($OD_{340}$) in a Synergy 4 microplate reader (BioTek Instruments) in 1 min intervals for 1 h at 37 °C. The initial absorbance in each well ($OD_{340}$ at $t = 0$) was set as the background, and was subtracted from $OD_{340}$ values of subsequent time points. The resulting $\Delta OD_{340}$ was blotted as a function of time. For dose–response analysis, values were normalized to the maximal change of $OD_{340}$ ($\Delta OD_{340\text{max}}$) of DMSO control experiments. $IC_{50}$ values of inhibitors were determined by fitting with GraphPad's normalized

nonlinear regression curve function with variable slope:

$$y = 100 / \left(1 + 10^{[(\text{Log IC}_{50} - \text{Log} x) \times \text{HillSlope}]}\right), \quad \text{where } x = \text{drug concentration}$$

Statistical significance for $OD_{340, \text{MAX}}$ was tested using non-parametric Kruksal–Wallis test (small sample number not allowing to test for Gaussian distribution using D'Agostino-Pearson omnibus normality test, Supplementary Fig. 2b).

**Preparation of proteins.** Bovine brain tubulin prepared according to well-established protocols[27] was purchased from the Centro de Investigaciones Biológicas (Microtubule Stabilizing Agents Group, fer@cib.csic.es), CSIC, Madrid, Spain. The recombinant expression and purification of the stathmin-like domain of RB3 and chicken TTL in bacteria, as well as the reconstitution of the T2R-TTL complex was performed as described previously[12–14]. The expression and purification of the PI3Kγ catalytic subunit (p110γ) was performed as described in previously[28].

**_In vitro_ PI3K inhibitor binding assay.** Dissociation constants ($K_d$) of compounds for p110γ, α, β, δ were determined using LanthaScreen technology (Life Technologies). First, dissociation constants of Kinase Tracer314 ($K_{d \text{ Tracer}}$, # PV6087) for p110γ (13.9 nM), p110α (2.2 nM), p110β (3.5 nM) and p110δ (4.1 nM) were determined following the manufacturer's instructions. Recombinant PI3Ks were prepared as $3 \times$ concentrated stocks of either N-terminally His-tagged p110α, p110β and p110δ associated with p85α or His-tagged p110γ (15 nM), biotin anti-His$_6$-tag antibody (6 nM, # PV6089) and LanthaScreen Eu-Streptavidin (6 nM, # PV5899) was prepared in $1 \times$ Kinase Buffer A (50 mM HEPES pH 7.5, 10 mM MgCl$_2$, 1 mM EGTA and 0.01% (v/v) Brij-35). $3 \times$ concentrated AlexaFluor647-labelled Kinase Tracer314 was prepared in Kinase Buffer A (Tracer314 concentrations for p110γ: 75 nM, p110α: 60 nM, p110β: 75 nM, p110δ: 30 nM). Compounds were fourfold serially diluted from 120 μM to 460 pM in kinase buffer A. Five microlitres of compound dilutions, 5 μl of $3 \times$ kinase/antibody mixture and 5 μl $3 \times$ Tracer314 solution were mixed in a 384-well plate (in duplicate) and incubated for 1 h at room temperature. Time-resolved FRET was measured with a Synergy 4 multi-mode microplate reader (Biotek Instruments; settings: 100 ms delay, 200 ms for data collection with 10 measurements per data point. Emission filters: 665/8 and 620/10 nm; Excitation filter: 340/30 nm; Dichroic mirror 400 nm).

For data analysis, mean background (wells with only kinase buffer A) was subtracted and the emission ratio was calculated by dividing the signal emitted at 665 nm from the acceptor (AlexaFluor647-labelled Tracer314) by the signal emitted at 620 nm from the donor (Eu-labelled antibody). $IC_{50}$ values were determined by plotting the remaining emission (in % of DMSO control) as a function of compound concentration (in logarithmic scale) and by fitting a normalized sigmoidal dose–response curve with variable slope

$$(y = 100 / (1 + 10^{[(\text{Log IC}_{50} - \text{Log} x) \times \text{HillSlope}]}))$$

to the data using GraphPad Prism. $K_d$ was calculated as

$$K_d = IC_{50} / (1 + ([\text{Tracer}] / K_{d \text{ Tracer}})$$

Experiments for p110α were repeated in three independent experiments in duplicate ($n = 3 \times 2$, Fig. 3d $n = 4 \times 2$–3). Inhibitor action on the other PI3K isoform was tested in two independent experiments in duplicate ($n = 2 \times 2$).

**PI3Kα activity determined by PtdIns(3,4,5)$P_3$ production.** PI3Kα-derived PtdIns(3,4,5)$P_3$ was detected using an AlphaScreen assay (K-1300) from Echelon Biosciences following the manufacturer's instructions. Inhibitors (stocks at 1.6 mM in DMSO) were diluted 10-fold in $1 \times$ KBZ (catalog number K-KBZ; Echelon Biosciences), followed by 10 steps of a four-fold serial dilution in $1 \times$ KBZ buffer supplemented with 10% DMSO. Recombinant human p110α/p85α PI3Kα complex (E-2000; Echelon Biosciences) was used, and the assay was set up in the following order: 2.5 μl of each inhibitor (concentration range of 40–0.00015 μM), 5 μl PI3Kα as supplied, and 2.5 μl 20 μM diC$_8$ PtdIns(4,5)$P_2$ (the short fatty acid chain PI3K substrate; P-4508; Echelon Biosciences) mixed with 40 μM ATP in $1 \times$ KBZ buffer. The reaction mixtures were incubated for 1 h at room temperature, before PtdIns(3,4,5)$P_3$ was detected using a AlphaScreen GST detection Kit (PerkinElmer) according to the supplier's instructions. Samples were finally processed on a Fusion Alpha plate reader (Packard). AlphaScreen signals (relative luminescent units, RLU) were depicted as a function of drug concentration. $IC_{50}$ values for PI3Kα inhibition were determined using GraphPad's 'nonlinear regression with variable slope' formula:

$$y = \text{bottom} + (\text{top} - \text{bottom}) / \left(1 + 10^{[(\text{Log EC}_{50} - \text{Log} x) \times \text{HillSlope}]}\right)$$

Where not indicated, inhibitor action was determined by duplicates of 10-step serial dilutions, and by quadruplicates for MTD265, PIKiN1-R1, PIKiN2-R1.

**Crystallization, data collection and structure solution.** Crystals of T$_2$R-TTL were grown as described in refs. 12, 13 and soaked overnight (BKM120, MTD147) or for 2 h (MTD265, MTD265-R1) in reservoir solutions containing 2 mM the indicated compounds. Crystals were fished directly from the drop and flash-cooled in liquid nitrogen. Standard data collection was performed at beamline X06DA at the Swiss Light Source (Paul Scherrer Institut, Villigen, Switzerland). Images were indexed and processed using XDS[29]. The structures were determined by the difference Fourier method in PHENIX[30] using the phases of the T$_2$R-TTL complex (PDB ID 4I4T) in the absence of ligands and solvent molecules as a starting point for refinement. The resulting model was further improved through iterative model rebuilding in Coot[31] and refinement in PHENIX[30]. Non-crystallographic symmetry restraints were applied in initial refinement stages and then omitted in the final cycles of refinement to account for structural variations between the non-crystallographic symmetry -related copies of α- and β-tubulin. Translation/Libration/Screw-refinement were included in the final cycles of refinement. Data collection and refinement statistics are given in Supplementary Table 7.

Figures were prepared using PyMOL (The PyMOL Molecular Graphics System, Version 1.5.0.5. Schrödinger, LLC). Chains in the T$_2$R-TTL complex were defined as follows: chain A, α-tubulin; chain B, β1-tubulin; chain C, α2-tubulin; chain D, β2-tubulin; chain E, RB3; chain F, TTL. Chains A and B were used throughout for the structural analyses and figure preparation.

Crystals of PI3Kγ were prepared as described in ref. 28. In short, crystals were grown using sitting-drop vapor diffusion by mixing 1 μl of PI3Kγ (4 mg ml$^{-1}$) and 1 μl of a reservoir solution (16–17% PEG 4000, 250 mM (NH$_4$)$_2$SO$_4$ and 100 mM Tris pH 7.5). Crystal seeds were introduced using the seed bead kit (Hampton Research). Individual soaking with each inhibitor generated ligand co-crystal structures. Three stocks of each inhibitor were made at concentrations of 10 μM, 100 μM and 1 mM in cryo buffer (23% PEG 4000, 250 mM (NH$_4$)$_2$SO$_4$, 100 mM Tris pH 7.5 and 14% glycerol). Crystals were incubated with increasing concentrations of inhibitors by adding 0.5 μl of inhibitor solution and allowing them to incubate for 60 min. As the last step, 1 μl was taken out of the drop and 1 μl of 1 mM inhibitor in cryo buffer was added and inhibitors were allowed to soak overnight. After soaking, the crystals were transferred to a fresh drop containing 1 mM inhibitor in cryo buffer and then immediately frozen by dunking the crystals in liquid nitrogen.

Diffraction data for the PI3K crystals were collected at 100 K at beamline I04 of the Diamond Light Source. Data were processed using XDS[29]. Phases were initially obtained by molecular replacement using Phaser[32], with the structure of PI3Kγ bound to BKM120 used as the search model (3SD5; ref. 3). Iterative model building and refinement were performed in COOT[31] and phenix.refine[30], with a final Rwork of 20.2 and an Rfree of 24.5 for the PI3Kγ structure bound to PIKiN2, and Rwork = 23.1 and Rfree = 27.7 for the PI3Kγ structure bound to PIKiN3. Refinement was carried out with rigid body refinement followed by Translation/Libration/Screw B-factor and xyz refinement. The final model was verified in Molprobity for the absence of both Ramachandran and Rotamer outliers[33]. Data collection and refinement statistics are shown in Supplementary Table 8.

**BMMC assays.** BMMCs were collected by centrifugation (160 g for 3 min), washed and starved in IL-3-free IMDM containing 2% heat-inactivated fetal calf serum, 2 mM L-glutamine and 1% penicillin–streptomycin for 1 h (0.5–1.0 × 10$^6$ cells per ml). Subsequently BMMC were treated for 30 min (37 °C, 5%CO$_2$) with PIKiN1 and PIKiN-R1 serial dilutions or DMSO and then stimulated with 2 μM adenosine (Ade) or 10 ng ml$^{-1}$ stem cell factor for 2 min at 37 °C. For western blot analysis, stimulation of cells was stopped on ice and cells were collected by centrifugation (16,000 g for 1 min at 4 °C), washed in 1 × PBS and lysed at 1 × 10$^7$ cells per ml in 2 × sample buffer (125 mM Tris-HCl (pH 6.8), 4% SDS, 10% β-ME, 20% glycerol, bromphenol blue). Proteins were denatured at 95 °C for 7 min and subjected to SDS–polyacrylamide gel electrophoresis (PAGE) and western blotting. Total PKB/Akt and pSer473 phosphorylated PKB/Akt were detected with specific antibodies (both from Cell Signaling Technologies, # 2920 and # 4058) and respective HRP-coupled secondary antibodies (Sigma # A4416 and # A6154). Enhanced chemiluminescence signals were detected on a Fusion FX-7 device (Vilber Lourmat). Signals were quantified using ImageJ. Phospho-Ser473 PKB signals were corrected for total PKB and plotted as % of stimulated controls as a function of drug concentration using GraphPad Prism as described above.

**Molecular dynamics calculations.** Protein and structured water coordinates were taken from the PIKiN3-p110γ structure, and BKM120 (from PDB ID 3SD5) was then docked in the two orientations shown in Fig. 3g,h. The two complexes were then submitted to dynamic/minimization calculations with liberated water molecules, but fixed ligand and protein coordinates.

All energy calculations and minimizations/dynamics used periodic boundary conditions and the YAMBER3 force field[34]. The system was then energy-minimized using steepest descent minimization, in order to remove conformational stress, followed by a simulated annealing minimization until convergence (< 0.05 kJ per mol per 200 steps). Two independent simulations were run: 1,000 fs at 1,000 K followed by 1,000 fs at 300 K. The complex was then minimized until convergence. Binding energies were calculated using the YASARA default.

**Synthetic chemistry.** See Supplementary Methods.

**Data availability.** The atomic coordinates and structure factors have been deposited in the Protein Data Bank (www.rcsb.org) under accession codes 5M7E (T2R-TTL-BKM120), 5M7G (T2R-TTL-MTD147), 5M8G (T2R-TTL-MTD265), 5M8D (T2R-TTL-MTD265-R1), 5JHA (p110γ-PIKiN2) and 5JHB (p110γ-PIKiN3). The data that support the findings of this study are available from the corresponding author upon request. BKM120 is commercially available from multiple suppliers. To obtain PQR309 please contact the corresponding author.

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

## Acknowledgements

We thank P. Hebeisen, B. Giese, A. Pfaltz, E. Jackson and J.B. Langlois for advice, discussion and synthesis of chemical precursors and compounds; M. Neubauer and D. Häusinger for chemical structure determination; Pascal Lorenz and the BioOptics Facility for support with microscopy; Robert Ivanek for bioinformatics support and R. Sriramaratnam for editorial help. We are grateful to G. Zaman and the NTRC team for help and expertise with high-content screening assays. This work was supported by the Swiss Commission for Technology and Innovation (CTI) by PFLS-LS grants 14032.1, 15811.2 and 17241.1; the Stiftung für Krebsbekämpfung grant 341, Swiss National Science Foundation grants 310030_153211 and 316030_133860 (to M.P.W.), and 310030B_138659 and 31003A_166608 (to M.O.S.); in part by European Union's Horizon 2020 research and innovation programme under the Marie Skłodowska-Curie grant agreement 675392, and grants BIO2013-42984-R (Ministerio de Economia

y Competitividad), S2010/BMD-2457 BIPEDD2 (Comunidad Autónoma de Madrid) to J.F.D.; and by the MRC to R.L.W. (U105184308).

## Author contributions

Authors contributed to the work as follows: (i) conception and design by T.B., A.E.P., J.E.B., D.F., M.O.S., M.P.W.; (ii) development of methodology by T.B., A.E.P., F.B., J.E.B., A.M., A.J.I., D.R., A.M.S., K.B., A.Ah., A.Ak., J.F.D., M.Z., R.L.W., M.O.S., M.P.W.; (iii) data acquisition by T.B., A.E.P., F.B., J.E.B., A.M., A.J.I., D.R., A.M.S., K.B., A.Ah., M.Z.; (iv) interpretation of data by T.B., A.E.P., J.E.B., A.M., A.Ah., A.Ak., J.F.D., D.F., M.Z., M.O.S., M.P.W.; (v) writing and revising of the manuscript by T.B.; A.E.P., J.E.B., D.R., A.Ak., M.O.S., M.P.W.; (vi) administrative, technical or material support by A.J.I., K.B., M.Z., M.P.W.; (vii) study supervision and planning by J.E.B., A.Ak., J.F.D., R.L.W., M.O.S., M.P.W. and (viii) structure based drug design by T.B., F.B., D.R., A.M.S., V.C., N.C., M.P.W.

## Additional information

**Competing financial interests:** T.B., A.E.P., J.E.B., A.M., A.J.I., D.R., A.M.S., K.B., A.Ah., A.Ak., J.F.D., M.Z., R.L.W. and M.O.S. have no conflicting interests. F.B., V.C., N.C. and D.F. are currently employees of PIQUR Therapeutics, and V.C., N.C., D.F. and M.P.W. own shares of PIQUR Therapeutics. All novel compounds mentioned in this work are in the public domain, except for BKM120 (Novartis) and PQR309 (licensed to PIQUR by University of Basel).

