## [Peer Review File · Nature Communications]

PEER REVIEW FILE

Reviewers' Comments:

Reviewer #1 (Remarks to the Author):

The study by Bohnacker et al. is an elegant demonstration that Buparlisib has distinct PI3K and microtubule inhibitory activities. The data are convincing, combining cell based and in vitro studies with crystallographic analysis of how the compounds interact with tubulin, and providing a structural basis for the observed activities. The data have clear clinical significance with regard to a drug that is already in the clinics, and the study has important implications for both PI3K and MT inhibitor design.

This reviewer has no significant criticisms. However, the title is not really appropriate – the key difference between BKM120 and MTD147 or PQR309 is not the one Dalton, but rather the replacement of the pyrimidine with pyridine or triazine. The fact that the molecular weights are similar is not the point.

Reviewer #2 (Remarks to the Author):

This is a revealing and high quality piece of research. The authors make a strong case that BKM120, a drug that entered clinical trials as a PI3K inhibitor also binds tubulin and appears to produce, at least a large part of, its major cellular and clinical impacts via this "off target" activity. The authors provide some structural data that backs-up this case and suggests some earlier interpretation of structural data examining interactions between BKM120 and PI3Ks was likely partly wrong. In developing the work a more evolved version of BKM120, PQR309, that does not inhibit microtubule function but is a potent PI3K inhibitor, has been developed and has entered clinical trials. It is unlikely that PQR309 will have no other off-target activities but that is not the subject of this study. Furthermore, there are existing PI3K inhibitors that do not interfere with tubulin function, as confirmed by the authors. BKM120 has been used in some studies as a PI3K inhibitor but it has not been widely accepted as a unique, gold-standard reference

compound in my opinion.

The paper is well written and the data are high quality, well presented and analysed and interpreted correctly. I do not have any issues that I believe the authors could readily address to make this paper more appropriate for publication in Nature Communications.

Reviewer #3 (Remarks to the Author):

This manuscript (NCOMMS-16-18629-T) entitled “One Dalton change in Buparlisib defines specific PI3K and tubulin inhibitors for therapeutic intervention” by Professor Wymann and colleagues presents a focused and interesting study designed to tweak out (investigate) the PI3K versus tubulin activity of BKM120. The manuscript is well-written with solid science supporting the conclusions. It is the recommendation of this reviewer that the manuscript be published after consideration of the points listed below:

1) Considering the myriad of chemical modifications that could be made to the core structure of BKM120, it would be instructive for the authors to comment a bit further on what guided their choice of central ring replacement (pyrimidine to pyridine) to generate a tubulin active agent void of PI3K active and conversely central ring replacement (pyrimidine to triazine) to reverse this activity profile. Presumably a series of structural modifications were considered and potentially additional compounds synthesized and evaluated? Additional commentary in this regard would be instructive.

2) The title, while likely to catch the attention (and wet the appetite) of readers, really should be revised. While stating that a 1 dalton change delineates tubulin versus kinase activity is technically correct [since replacement of a nitrogen atom (atomic weight 14) with a CH group (atomic weight $12+1 = 13$) results in a 1 dalton change], it is not the weight change but rather most likely the actual atom change issue and the resultant electrostatic, steric (possibly), and related interactions with the biological targets that delineate the separation of activity through these analogues of BKM120. A suggestion is to state “replacement of a nitrogen atom with a CH group”

3) The authors may wish to comment a bit further on the potential benefits of a dual-mode PI3K/inhibitor of tubulin polymerization as an anticancer agent versus a clean P13K agent. The authors suggest that removing the tubulin component may reduce side-effects, etc., but what are the benefits of a dual-mode inhibitor? Are there other P13K inhibitors in clinical development that are structurally similar to BKM120? If so, have these other agents been evaluated for dual activity? It seems important to consider and speculate (perhaps a bit) on the perceived benefits and downside to dual-activity inhibitors of this type. The authors address portions of this briefly.

4) A potential conflict of interest in regard to a connection of several co-authors with PIQUR Therapeutics is noted in the “Conflicts of Interest” section. Is this conflict related to certain of the inhibitors presented in the paper? Please clarify in the text if this is indeed the case by perhaps mentioning which are related to PIQUR.

5) What was the rationale for selecting the structural modifications that led to PIKiN1, 2 and their regioisomers? In other words, of the many modifications that could be made to reduce symmetry and thus explore rotational orientations, why were piperidine and chloromethyl-azetidine-methanol substitutions utilized? The authors mention a hydrogen bond consideration, but were other analogues prepared as well and these provide the most efficacious? Some further clarification would be useful.

6) The synthetic chemistry is presented in the Supporting Information file. It would be helpful if the authors added the amount and mmoles to the final yield presented for each synthesized compound. While the reader can back-calculate this from the yield, it would be helpful to have this information directly presented. It would also be helpful if the authors would add a reference citation(s) for each intermediate and target final compound that has previously been described in the literature.

7) The docx versions of the manuscript and Supporting file that open on this reviewer’s computer have several instances where a square box appears – likely where a symbol font (or related) was entered by the authors. This is a computer to computer manifestation and is only pointed out by this reviewer as an encouragement for the authors to double-check these fonts and verify that they are correct (and transfer correctly) to the final print version of the article.

8) Although unavoidable to a certain extent, this manuscript describes a variety of compounds by a variety of compound codes as well as other names (nocodazole and colchicine, for example) and since the compounds are not all shown structurally in one table or figure, the reader ends up going from figure to figure to find the corresponding structures. In addition, the structures of certain of the reference compounds are not shown in the manuscript (GDC0941 and 980, etc). If possible, it might be useful to add a figure to the main manuscript (or one of the supporting files) in which all of the structures that are mentioned in the manuscript are shown along with their identifying number and/or name.

9) Fig 1C and 1D the type of assay used to determine these data should be mentioned in the figure or figure caption. Presumably in triplicate or greater but this should be mentioned as well.

10) Did the authors carry out a standard cell-free assay to determine an IC₅₀ for inhibition of tubulin polymerization in comparison to colchicine? Similar to the work carried out by Ernie Hamel at the US National Cancer Institute and others? The authors carry out a microtubule reconstitution assay and an assay in HeLa cells. Also, since their inhibitor-tubulin crystallography indicates binding at the colchicine site, it might be instructive to carry out a competitive binding assay with radiolabeled colchicine to determine the ability of these inhibitors to effectively compete for that site. Kinetics of binding are likely important and may differ substantially from colchicine itself.

Reviewer #4 (Remarks to the Author):

The manuscript by Wymann et al. described the design of selective PI3K inhibitor PQR309 and tubulin inhibitor MTD147 based on replacement of only one atom in the core pyrimidine ring of BKM120. An extensive battery of biochemical, compounds cellular growth arrest phenotypes and microtubule dynamics studies show that the antiproliferative activity of BKM120 is rooted in the inhibition of tubulin rather than PI3K. Furthermore, the co-crystal structures of tubulin and PI3K in complex with BKM120 and its derivatives were determined to uncover the molecular underpinnings of the selectivity of BKM120 series to tubulin and PI3K.

Overall, the article is based on solid experiments that support the authors' conclusions. This finding will revise our previous notion pertaining to the mechanism of drug action of BKM120 that the antiproliferative action of BKM120 occurs through the inhibition of PI3K. Some following concerns about this work could improve this manuscript if the authors can solve them in the revision.

1. BKM120 is a microtubule-destabilizing agent and it bound between the α - and β -tubulin subunits inhibiting 'curved-to-straight' conformational change. Does BKM120 interfere with microtubule polymerization (abstract) or induce microtubule depolymerization (268-269)? Is there any evidence?

2. The high affinity binding of BKM120 to tubulin occurs via the core C-H group oriented towards β Met259, is there an Met259 mutant experiment to prove it? The same experiment should be done to prove the high affinity binding of PQR309 to PI3K.

3. The authors found that MTD147 ($\geq 0.5 \mu\text{M}$) and BKM120 ($\geq 1 \mu\text{M}$), but not PQR309 ($5 \mu\text{M}$), attenuated microtubule growth rates and increased catastrophe frequency in vitro (Fig. 2a-c) and in cells (Fig. 2d-f). Fig 2a and 2d is indistinct and the results are not obvious. The size of control and other pictures don't match perfectly.

4. PI3K/AKT signaling pathway plays a critical role in cell growth, proliferation, and differentiation. PI3Ks are well-known cancer driver genes. Inhibition of PI3K isoforms is capable of inhibiting cancer cells proliferation. Indeed, BKM120 can bind to the ATP-binding site of all PI3K isoforms with high binding affinities. In the paper the authors stated that the antiproliferative action of BKM120 is not through PI3Ks inhibition. Thus, I have a puzzling that what's the biological function of BKM120 bound to PI3Ks.

5. The two derivatives of MTD147, MTD265 and MTD265-R1, showed different binding affinities to tubulin. Based on the co-crystal structures of tubulin-MTD265 and tubulin-MTD265-R1 complexes, the major interactions between tubulin and MTD265, MTD265-R1 in the active site are very similar. The authors stated that Met259 from beta-tubulin is responsible for the higher affinity of MTD265 compared to MTD265-R1, owing to the formation of additional weak interaction between pyrimidine C-H group of MTD265 and the sulfur atom of Met259 from beta-tubulin. To support this notion, the authors need to do further experimental tests via mutation of the Met259 from beta-tubulin to alanine. How is the change of the binding

affinities of MTD265 and MTD265-R1 to M259A tubulin.

6. The electron density of BKM120 in the crystal structure of PI3K γ -BKM120 complex is not clearly seen. The possible reason is due to the weaker binding affinity of BKM120 to PI3K γ compared to alpha, beta, and delta PI3K isoforms. Indeed, the binding affinity of BKM120 to PI3K γ is about 10-20 fold weaker compared to alpha, beta, and delta PI3K isoforms. Thus, the authors can attempt to determine the co-crystal structures of BKM120 bound to alpha, beta, and delta PI3K isoforms.

7. In the study of the selectivity of BKM120 regioisomer and derivatives to PI3Ks, the authors stated that several structural water molecules that take part in the water-mediated hydrogen binding interactions between D964, D836 of PI3K γ and the core pyrimidine nitrogen atom of BKM120 are responsible for the higher affinity of BKM120 to PI3K compared to tubulin. This conclusion is hypothesized without any biological experimental data. To support the author's notion, more biological experiments such as the binding affinities of BKM120 to D964A, D836A of PI3K γ mutants should be performed. The D964A, D836A of PI3K γ mutants will disrupt the previous formation of water-mediated hydrogen binding interactions between D964, D836 of PI3K γ and BKM120

Reviewer #5 (Remarks to the Author):

The authors present chemical, structural and biological data to decode the inhibition pattern of BKM120, a compound that crosses the bbb. They demonstrate the changes in the pyrimidine core of BKM120 that allow to deconvolute the inhibition of PI3K vs. the microtubule depolarization activity (MDA) by synthesis of BKM120 derivatives and their testing in vitro, in cell assays and x-ray crystallography. Specifically, they show that BKM120 has an off target effect that stops cell proliferation by mitotic block instead of PI3K inhibition. They show in vitro and in cells that BKM120 attenuates microtubule growth rate and has increase catastrophe activity. On the other hand PIQ309 is specific PI3K inhibitor with no MDA activity. The authors combine the biological activity and x-ray structure analysis of 2 regioisomers (derivatives of BKM120) to clearly establish the orientation of the core pyrimidine in tubulin complexes. Furthermore, they determined the structure of BKM120, MTD147, MTD265-R1 and MTD265 with tubulin, as well as the structure of PI3K γ with PIK γ 2 and PIK γ 3. Interestingly, the presence of a nitrogen atom in the triazine core of PIK309, prevents binding of PIQ309 to tubulin.

Although the manuscript is a well written and convincing story of how to fix off target effects BKM120 since the core nitrogen atoms define the PI3K interaction. The crystallographic work requires more work before publication

1- the software packages used for data processing of the x-ray data and for refinement should be

clearly stated not with a reference.

2- The complex structures of PI3K γ -PIKIN2 and PIKIN3 geometrical parameters are too tight. The bond length and bond angle reported are 0.003 and \sim 0.5. These values strongly suggest that loosening the geometrical restraint is required to re-refine before the manuscript is published since they are a lot smaller than the commonly accepted values reported by Engh & Huber (bond length \sim 0.018)

3- The fact that the solvent' B-factor (74) is smaller than that of the protein (100) is weird. DO the authors mean protein for complex? In structure PI3K γ -PIKIN2.

4- The fact that the solvent and inhibitor have smaller b-factor (by a lot) than the 'complex' in such a large structure is concerning. This should be fixed and explained.

5- The complexes of Tubulin with all 4 inhibitors have target of bond length and angles that do not deviate enough from ideality (0.008). Please, re refine with more realistic targets

It is absolutely require that the PDB ID codes are included in the manuscript before it is accepted for publication.

The biological experiments are reported as results of 3 independent experiments as usually required.

Typos and minor correction

Supplementary table 6

Add commas between cell dimensions to match style of table 5;

Reviewers' comments:

AU - authors' comments and replies to queries

Reviewer #1 (*Remarks to the Author*):

The study by Bohnacker et al. is an elegant demonstration that Buparlisib has distinct PI3K and microtubule inhibitory activities. The data are convincing, combining cell based and in vitro studies with crystallographic analysis of how the compounds interact with tubulin, and providing a structural basis for the observed activities. The data have clear clinical significance with regard to a drug that is already in the clinics, and the study has important implications for both PI3K and MT inhibitor design.

This reviewer has no significant criticisms. However, the title is not really appropriate – the key difference between BKM120 and MTD147 or PQR309 is not the one Dalton, but rather the replacement of the pyrimidine with pyridine or triazine. The fact that the molecular weights are similar is not the point.

AU: We thank this reviewer for the appreciation of our work. His comments concerning the title are correct, but it was an attempt to communicate to non-chemists that closely related molecules can affect proliferation and cell cycle very differently. As requested, the title has been changed to "Deconvolution of Buparlisib's mechanism of action defines specific PI3K and tubulin inhibitors for therapeutic intervention".

Reviewer #2 (*Remarks to the Author*):

This is a revealing and high quality piece of research. The authors make a strong case that BKM120, a drug that entered clinical trials as a PI3K inhibitor also binds tubulin and appears to produce, at least a large part of, its major cellular and clinical impacts via this "off target" activity. The authors provide some structural data that backs-up this case and suggests some earlier interpretation of structural data examining interactions between BKM120 and PI3Ks was likely partly wrong. In developing the work a more evolved version of BKM120, PQR309, that does not inhibit microtubule function but is a potent PI3K inhibitor, has been developed and has entered clinical trials. It is unlikely that PQR309 will have no other off-target activities but that is not the subject of this study. Furthermore, there are existing PI3K inhibitors that do not interfere with tubulin function, as confirmed by the authors. BKM120 has been used in some studies as a PI3K inhibitor but it has not been widely accepted as a unique, gold-standard reference compound in my opinion. The paper is well written and the data are high quality, well presented and analysed and interpreted correctly. I do not have any issues that I believe the authors could readily address to make this paper more appropriate for publication in Nature Communications.

AU: We thank this reviewer for his kind comments and compliments on our work's quality.

Reviewer #3 (Remarks to the Author):

This manuscript (NCOMMS-16-18629-T) entitled "One Dalton change in Buparlisib defines specific PI3K and tubulin inhibitors for therapeutic intervention" by Professor Wymann and colleagues presents a focused and interesting study designed to tweak out (investigate) the PI3K versus tubulin activity of BKM120. The manuscript is well-written with solid science supporting the conclusions. It is the recommendation of this reviewer that the manuscript be published after consideration of the points listed below:

1) Considering the myriad of chemical modifications that could be made to the core structure of BKM120, it would be instructive for the authors to comment a bit further on what guided their choice of central ring replacement (pyrimidine to pyridine) to generate a tubulin active agent void of PI3K active and conversely central ring replacement (pyrimidine to triazine) to reverse this activity profile. Presumably a series of structural modifications were considered and potentially additional compounds synthesized and evaluated? Additional commentary in this regard would be instructive.

AU-1: There are indeed many possibilities to enforce or attenuate PI3K and/or tubulin binding of BKM120 derivatives. MTD265 is a nice example how the substitution of a morpholino group can dramatically shift activity towards MT disruption. We have indeed synthesized a large variety of PI3K inhibitors and MT active compounds. The main scope of this work was to dissect the action of BKM120: the approach and selection of compounds presented in the current work was thus driven by the search for molecules best suited to evaluate and segregate the inherent activities of BKM120. Here, the replacement of the core pyrimidine provided a unique opportunity to maintain (known) PI3K interactions and retain physicochemical properties of BKM120 derivatives. We would not like to add additional compounds, and not mention them explicitly, as we are currently preparing follow-up papers on BKM120-derived MT disruptors with novel properties, as well as PI3K inhibitors with substituted morpholino groups, which will refer to the work presented here. We have added some explanatory text elaborating the choice for MT vs. PI3K SAR compounds in very general terms.

2) The title, while likely to catch the attention (and wet the appetite) of readers, really should be revised. While stating that a 1 dalton change delineates tubulin versus kinase activity is technically correct [since replacement of a nitrogen atom (atomic weight 14) with a CH group (atomic weight 12+1 = 13) results in a 1 dalton change], it is not the weight change but rather most likely the actual atom change issue and the resultant electrostatic, steric (possibly), and related interactions with the biological targets that delineate the separation of activity through these analogues of BKM120. A suggestion is to state "replacement of a nitrogen atom with a CH group"

AU-2: As suggested, the title has been changed to "'Deconvolution of Buparlisib's mechanism of action defines specific PI3K and tubulin inhibitors for therapeutic intervention".

3) The authors may wish to comment a bit further on the potential benefits of a dual-mode

P13K/inhibitor of tubulin polymerization as an anticancer agent versus a clean P13K agent. The authors suggest that removing the tubulin component may reduce side-effects, etc., but what are the benefits of a dual-mode inhibitor? Are there other P13K inhibitors in clinical development that are structurally similar to BKM120? If so, have these other agents been evaluated for dual activity? It seems important to consider and speculate (perhaps a bit) on the perceived benefits and downside to dual-activity inhibitors of this type. The authors address portions of this briefly.

AU-3: We have added a section discussing recent results concerning BKM120 and GDC0941 studies in combination with paclitaxel. We feel that a discussion of dual-mode vs. single-mode inhibitors in the given context is presently highly speculative, and that statements in favor of PQR309 must be carefully weighed against a conflict of interest.

4) A potential conflict of interest in regard to a connection of several co-authors with PIQUR Therapeutics is noted in the "Conflicts of Interest" section. Is this conflict related to certain of the inhibitors presented in the paper? Please clarify in the text if this is indeed the case by perhaps mentioning which are related to PIQUR.

AU-4: All compounds mentioned in the work, except for BKM120 (Novartis) and PQR309 (PIQUR) are in the public domain. A short statement has been added to the conflict of interest section.

5) What was the rationale for selecting the structural modifications that led to PIKiN1, 2 and their regioisomers? In other words, of the many modifications that could be made to reduce symmetry and thus explore rotational orientations, why were piperidine and chloromethyl-azetidine-methanol substitutions utilized? The authors mention a hydrogen bond consideration, but were other analogues prepared as well and these provide the most efficacious? Some further clarification would be useful.

AU-5: PIKiN1 and PIKiN1-R1 are the simplest deviations from BKM120 to generate asymmetric molecules with only one possibility to form a backbone hydrogen bond interaction with Val882. We have produced other regioisomeric pairs with a core pyrimidine, yielding the same results (e.g. spiro-, thiomorpholino, etc, substituted compounds). The PIKiN1/-R1 pair is the simplest set of compounds to explain the importance of the pyrimidine core orientation. As the piperidine group cannot be readily distinguished from a morpholino substitution in our crystal structures, the chloromethyl-azetidine-methanol substitution was introduced to provide an electron dense atom (chlorine). Many other substitutions could be selected, but we choose piperidine and chloromethyl-azetidine-methanol substitutions for the sake of a simple argumentation and technical reasons (structure determination). The newly introduced Supplementary Tables 4 and 6 should now provide a better overview, and also illustrates that the MTD265/R1 pair follows the same pattern. The text was changed to better clarify the selection of compounds.

6) *The synthetic chemistry is presented in the Supporting Information file. It would be helpful if the authors added the amount and mmoles to the final yield presented for each synthesized compound. While the reader can back-calculate this from the yield, it would be helpful to have this information directly presented. It would also be helpful if the authors would add a reference citation(s) for each intermediate and target final compound that has previously been described in the literature.*

AU-6: The amounts of compounds produced have been added. For most compounds, no previous synthesis has been reported (patent literature not included). Missing references have been added.

7) *The docx versions of the manuscript and Supporting file that open on this reviewer's computer have several instances where a square box appears – likely where a symbol font (or related) was entered by the authors. This is a computer to computer manifestation and is only pointed out by this reviewer as an encouragement for the authors to double-check these fonts and verify that they are correct (and transfer correctly) to the final print version of the article.*

AU-7: This seems to be a Mac/PC & MS-Word 2011 problem. Using MS-Word 2016 eliminated these "invisible" characters.

8) *Although unavoidable to a certain extent, this manuscript describes a variety of compounds by a variety of compound codes as well as other names (nocodazole and colchicine, for example) and since the compounds are not all shown structurally in one Table or figure, the reader ends up going from figure to figure to find the corresponding structures. In addition, the structures of certain of the reference compounds are not shown in the manuscript (GDC0941 and 980, etc). If possible, it might be useful to add a figure to the main manuscript (or one of the supporting files) in which all of the structures that are mentioned in the manuscript are shown along with their identifying number and/or name.*

AU-8: As suggested, a compound overview has been added in Supplementary Fig. 1p. Figure legend 1 refers to this formula sheet.

9) *Fig 1C and 1D the type of assay used to determine these data should be mentioned in the figure or figure caption. Presumably in triplicate or greater but this should be mentioned as well.*

AU-9: The figure legend has been adapted, and now clarifies the assay and mentions that a "9-point serial drug dilution" was performed, as well as statistical methods applied.

10a) *Did the authors carry out a standard cell-free assay to determine an IC_{50} for inhibition of tubulin polymerization in comparison to colchicine? Similar to the work carried out by Ernie Hamel at the US National Cancer Institute and others?*

AU-10a: The requested data has been added in Supplementary Fig. 2a-g; Supplementary

Fig. 4c-d, and j; and has been summarized in Supplementary Table 4. Included were PQR309, GDC0941, GDC0980, BKM120/R1, MTD147, MTD265/R1 and nocodazole. Colchicine was not included in the calculation of an IC_{50} using the MT turbidity assay, as its action is temperature sensitive and its binding quasi irreversible.

10b) The authors carry out a microtubule reconstitution assay and an assay in HeLa cells. Also, since their inhibitor-tubulin crystallography indicates binding at the colchicine site, it might be instructive to carry out a competitive binding assay with radiolabeled colchicine to determine the ability of these inhibitors to effectively compete for that site. Kinetics of binding are likely important and may differ substantially from colchicine itself.

AU-10b: As pointed out above, colchicine binding to tubulin is very temperature-sensitive and colchicine is "locked" into the binding pocket. Competition experiments have therefore often failed for colchicine, although they were attempted with validated "colchicine-binding site"-binders. As shown in Brachmann et al. (2012, Mol Cancer Ther, 11, 1747-57), the approach already failed for BKM120. We agree that kinetic parameters are an important feature for molecules targeting MTs. We have performed competition experiments with the bicyclic colchicine analogue 2-methoxy-5-(2',3',4'-trimethoxyphenyl)-2,4,6-cycloheptatrien-1-one (MTC), and have determined preliminary association constants in the range of $0.3 \pm 0.1 \times 10^5 \text{ M}^{-1}$ for BKM120 (information for reviewer). Further studies will be presented in a follow-up SAR paper dealing with BKM120-derived MT binders emerging from MTD265. The compounds contained in this future work confirm that the colchicine-binding site is the target site for BKM120-derived MT blockers.

Reviewer #4 (Remarks to the Author):

The manuscript by Wymann et al. described the design of selective PI3K inhibitor PQR309 and tubulin inhibitor MTD147 based on replacement of only one atom in the core pyrimidine ring of BKM120. An extensive battery of biochemical, compounds cellular growth arrest phenotypes and microtubule dynamics studies show that the antiproliferative activity of BKM120 is rooted in the inhibition of tubulin rather than PI3K. Furthermore, the co-crystal structures of tubulin and PI3K in complex with BKM120 and its derivatives were determined to uncover the molecular underpinnings of the selectivity of BKM120 series to tubulin and PI3K.

Overall, the article is based on solid experiments that support the authors' conclusions. This finding will revise our previous notion pertaining to the mechanism of drug action of BKM120 that the antiproliferative action of BKM120 occurs through the inhibition of PI3K. Some following concerns about this work could improve this manuscript if the authors can solve them in the revision.

1. BKM120 is a microtubule-destabilizing agent and it bound between the α - and β -tubulin ubunits inhibitting 'curved-to-straight' conformational change. Dose BKM120 interfere with microtubule polymerization(abstract) or induce microtubule depolymerization(268-269)? Is there any evidence?

AU-1: Mechanistically, BKM120 binds to free tubulin or tubulin at the plus end of an MT (curved confirmation needed to provide "open" binding pocket, see Fig. 3). BKM120's primary function is therefore to prevent MT polymerization. We have also added classical tubulin polymerization assays to visualize this in the new Supplementary Fig. 2. Fig. 2 provides a more sophisticated approach to demonstrate the same. When tubulin - due to BKM120-binding and stabilization of the curved conformation- cannot integrate into growing MTs, this leads in the long run to MT depolymerization. We have adapted the text criticized by the reviewer to state this more carefully.

2a. The high affinity binding of BKM120 to tubulin occurs via the core C-H group oriented towards β Met259, is there an Met259 mutant experiment to prove it?

AU-2a: We provide an extensive chemical SAR study, biochemical, structural and cellular experiments illustrating differential binding activities for regioisomers. Met259 is only one point of interaction. See text and legend to Fig. 3 for more residues in close proximity of the ligands (Cys241, Leu248, Ala250, Ala316, Ile318, Ala354, etc.). For additional information/discussion, see our reply to query #5.

2b. The same experiment should be done to prove the high affinity bingding of PQR309 to PI3K..

AU-2: We are not sure what the reviewer refers to. There is no Met259 in PI3K, and inhibitor-PI3K interaction is defined by mostly electrostatic interactions.

3. The authors found that MTD147 ($\geq 0.5 \mu\text{M}$) and BKM120 ($\geq 1 \mu\text{M}$), but not PQR309 ($5 \mu\text{M}$), attenuated microtubule growth rates and increased catastrophe frequency in vitro (Fig. 2a-c) and in cells (Fig. 2d-f). Fig 2a and 2d is indistinct and the results are not obvious. The size of control and other pictures don't match perfectly.

AU-3: Fig. 2a and d are in excellent agreement with the quantification data shown. Images represent time-laps of microtubules visualized with GFP-tagged EB3, and are so-called kymographs. While the controls (DMSO) show long microtubules, BKM120 and MTDs shorten the mean length of microtubules. To save space, images were truncated on the right in a way to not cut away any microtubules. Where microtubules were depolymerized, the image width could be reduced more. We have adapted the figure legend to point to this fact.

4. PI3K/AKT signaling pathway plays a critical role in cell growth, proliferation, and differentiation. PI3Ks are well-known cancer driver genes. Inhibition of PI3K isoforms is capable of inhibiting cancer cells proliferation. Indeed, BKM120 can bind to the ATP-binding site of all PI3K isoforms with high binding affinities. In the paper the authors stated that the antiproliferative action of BKM120 is not through PI3Ks inhibition. Thus, I have a puzzling that what's the biological function of BKM120 bound to PI3Ks.

AU-4: As illustrated in detail in the manuscript, BKM120 is a PI3K inhibitor, but its dominant action on cellular growth is exerted by its effects on microtubule dynamics and the resulting mitotic arrest. The comparison with MTD147 makes the point that MTD147 (a "BKM120 with very little PI3K activity") produces the same cellular phenotype as BKM120. Concerning cell cycle arrest, the BKM120-mediated inhibition of PI3K seems therefore to be negligible.

5a. The two derivatives of MTD147, MTD265 and MTD265-R1, showed different binding affinities to tubulin. Based on the co-crystal structures of tubulin-MTD265 and tubulin-MTD265-R1 complexes, the major interactions between tubulin and MTD265, MTD265-R1 in the active site are very similar.

AU-5a: MTD265 and MTD265-R1 are derivatives of BKM120 (pyrimidine core); MTD147 has a symmetry axis/plane and has two indistinguishable ways to bind to tubulin. Regioisomeric derivatives of MTD147 do not exist if replacing one morpholine. We clearly demonstrate with chemical SAR studies and biochemical experiments that interactions of MTD265 and MTD265-R1 with tubulin are very different, which results in a >30x fold change in biological activity. What is similar is the orientation and placement of the morpholino and pyrrolidine groups as demonstrated by the presented high resolution X-ray crystal structures. This defines the orientation of the core pyrimidine, and defines multiple interactions, including the one with Met259. The proximities of the tubulin side chains and MTD265 and MTD265-R1 are indeed very similar, but not the resulting binding energies: here hydrophobic interactions for MTD265 are replaced by repulsions for MTD265-R1. This is also illustrated by the higher mobility of the Lys352 side chain.

5b. The authors stated that Met259 from beta-tubulin is responsible for the higher affinity of MTD265 compared to MTD265-R1, owing to the formation of additional weak interaction between pyrimidine C-H group of MTD265 and the sulfur atom of Met259 from beta-tubulin.

AU-5b: As the reviewer points out, the proposed MTD265 pyrimidine C-H Met259 interaction is a hydrophobic, weak type interaction (Bissantz 2010, J Med Chem, 53, 5061-84). The Met259 interaction contributes, but is not the only "responsible" interaction for high-affinity MT binding.

5c. To support this notion, the authors need to do further experimental tests via mutation of the Met259 from beta-tubulin to alanine. How is the change of the binding affinities of MTD265 and MTD265-R1 to M259A tubulin.

AU-5c: Besides unsurmountable technical difficulties to produce recombinant, mutated tubulin heterodimers, a M259A mutation would only replace one hydrophobic interaction with another. Moreover, the Met259 interaction contributes to a differential MTD265 (C-H attraction) and MTD265-R1 (N lone pair repulsion) binding, but is - as described in the manuscript in detail - only one of many hydrophobic interactions. We therefore do not think that the proposed experiment will produce conclusive results.

6. The electron density of BKM120 in the crystal structure of PI3Kgama-BMK120 complex is not clearly seen. The possible reason is due to the weaker binding affinity of BKM120 to PI3Kgama compared to alpha, beta, and delta PI3K isoforms. Indeed, the binding affinity of BKM120 to PI3Kgama is about 10-20 fold weaker compared to alpha, beta, and delta PI3K isoforms. Thus, the authors can attempt to determine the co-crystal structures of BKM120 bound to alpha, beta, and delta PI3K isoforms.

AU-6: We are not sure what the referee refers to. We did not show a PI3K γ -BKM120 complex structure. All presented structural data show excellent resolution and electron densities. Crystallization and structural analysis of other PI3K isoforms is out of scope for this work.

7a. In the study of the selectivity of BKM120 regioisomer and derivatives to PI3Ks, the authors stated that several structural water molecules that take part in the water-mediated hydrogen binding interactions between D964, D836 of PI3Kgama and the core pyrimidine nitrogen atom of BKM120 are responsible for the higher affinity of BKM120 to PI3K compared to tubulin.

AU-7a: This statement is not entirely correct; it does not reflect the manuscript's text. While the interaction of BKM120 and tubulin is dominated by hydrophobic interactions, the inhibitor-PI3K interaction displays some important electrostatic and a number of H-bond interactions. The different binding constants of PI3K inhibitor regioisomers are a fact. The resolved water molecules have been identified in several crystal structures, and provide a valuable explanation for the preferred binding of one PI3K inhibitor regioisomer over the other (the PIKiNx vs. PIKiNx-R1 pairs and MTD265-R1 vs MTD265). The main point of the water molecule interacting with the core N of pyrimidine and triazine

containing molecules is thus that it provides an explanation for the observed and clearly proven preferred orientation. Molecular dynamics calculations support this notion.

7b. This conclusion is hypothesized without any biological experimental data. To support the author's notion, more biological experiments such as the binding affinities of BKM120 to D964A, D836A of PI3K γ mutants should be performed. The D964A, D836A of PI3K γ mutants will disrupt the previous formation of water-mediated hydrogen binding interactions between D964, D836 of PI3K γ and BKM120

AU-7b: As mentioned above, we have compelling evidence for PI3K inhibitor binding affinities and the molecules' orientation. There is a lot of structural and biochemical evidence presented to support our case. We have now added an overview panel of chemical compounds (Supplementary Fig. 1p) and activities (Supplementary Tables 4 and 6) to simplify navigation. The suggestion to mutate the Asp residues seems tempting, but does not yield novel insight: as depicted in Fig. 4.1, D964 and D836 do not only interact with the inhibitors but also with ATP. We know from previously published studies (Walker et al., Mol Cell, 2000), that these residues are required to maintain an intact, functional catalytic pocket in all PI3Ks and are maximally conserved residues. Mutation of either residue to Ala renders PI3K inactive (loss of ATP-binding; D964 is in the DFG motif). These mutations do not only disturb the water network, but they also eliminate important electrostatic and H-bond interactions. An expected drop in BKM120 affinity would be unlikely to be linked to the loss of the water network alone. We would like to argue, that our chemical, biochemical and structural data provides a much more elegant approach to investigate core orientation-dependent binding.

Figure 4.1 for reviewers. The above histogram displays the buried surface area and points of proximity of p110 γ (the PI3K γ catalytic subunit) and bound ATP (red, from PDB ID 1E8X) or PI3K inhibitor PIKIN2.

Reviewer #5 (Remarks to the Author AU):

The authors present chemical, structural and biological data to decode the inhibition pattern of BKM120, a compound that crosses the bbb. They demonstrate the changes in the pyrimidine core of BKM120 that allow to deconvolute the inhibition of PI3K vs. the microtubule depolarization activity (MDA) by synthesis of BKM120 derivatives and their testing in vitro, in cell assays and x-ray crystallography. Specifically, they show that BKM120 has an off target effect that stops cell proliferation by mitotic block instead of PI3K inhibition. They show in vitro and in cells that BKM120 attenuates microtubule growth rate and has increase catastrophe activity. On the other hand PIQ309 is specific PI3K inhibitor with no MDA activity. The authors combine the biological activity and x-ray structure analysis of 2 regioisomers (derivatives of BKM120) to clearly establish the orientation of the core pyrimidine in tubulin complexes. Furthermore, they determined the structure of BKM120, MTD147, MTD265-R1 and MTD265 with tubulin, as well as the structure of PI3Kg with PIKiN2 and PIKiN3. Interestingly, the presence of a nitrogen atom in the triazine core of PIK309, prevents binding of PQR309 to tubulin.

Although the manuscript is a well written and convincing story of how to fix off target effects BKM120 since the core nitrogen atoms define the PI3K interaction. The crystallographic work requires more work before publication

AU: We appreciate the comment on the manuscript being well written, and have added additional details on the crystallographic work as described point by point below.

1- the software packages used for data processing of the x-ray data and for refinement should be clearly stated not with a reference.

AU-1: We apologize for the oversight of the absence of the crystallographic collection, processing, and refinement methodology. Extended paragraphs have been added to the methods section explaining these details in the section “Crystallization, data collection and structure solution” (both for tubulin and PI3K γ structural analysis).

2- The complex structures of PI3Kg-PIKiN2 and PIKiN3 geometrical parameters are too tight. The bond length and bond angle reported are 0.003 and $\sim 0.5^\circ$. These values strongly suggest that loosening the geometrical restraint is required to re-refine before the manuscript is published since they are a lot smaller than the commonly accepted values reported by Engh & Huber (bond length ~ 0.018)

AU-2: Refinement was carried out in phenix.refine, and essential to refinement was the optimization of the X-ray stereochemistry weighting. We used the standard default weightings within Phenix.refine, and this was essential to maintain the absence of Ramachandran and rotamer outliers.

Refinement carried out in the absence of X-ray stereochemistry weighting was performed at the suggestion of the reviewer leading to bond lengths rmsd of 0.022 and angles rmsd of 1.99 for the PIKiN2. However, this led to significantly worse overall statistics as rfree rose to 0.27, with 2.2% Ramachandran outliers, and 4.0% rotamer outliers for the PIKiN2 structure. Re-refinement was also performed by varying the X-ray stereochemistry weighting in phenix.refine (wxc scale) from the default value of 0.5. Refinement was performed with wxc scale values of 0.75, 1.0, and 1.5 leading to loosened geometrical restraints. All changes in weighting did slightly increase bond rmsd to a maximum of 0.007, but led to an increased Rfree for all structures, and an increase in both Ramachandran and Rotamer outliers.

Relevant to this discussion is the statement from the Phenix developers (as given in the FAQs): “[The observation that the RMS (bonds) and RMS (angles) in the output model are too low after weight optimization] is a common misconception about geometry deviations, based partly on anecdotal experience with other refinement programs. The target RMSDs come from looking at very accurate high-resolution small molecule structures, so they reflect the real variation that should occur in geometry. At lower resolution, even though you know that the bond lengths and bond angles must vary as much as they do in high-resolution structures, there isn’t enough experimental data to tell in which direction they should deviate from the expected values. So it is reasonable for the refined RMSDs to be lower than the targets. (Anecdotally, we have found that phenix.refine often refines to much tighter RMSDs with similar R-frees to other programs. This may reflect different approaches used in the geometry restraints, X-ray target, or optimization methods, but it should not be cause for concern.)” (<https://www.phenix-online.org/documentation/faqs/refine.html>)

For these reasons, we feel that our original refinement strategy was optimal and the bond length and angle rmsds are consistent with similar resolution structures refined in Phenix.refine previously published in Nature Communications (examples PDB: 5FI9, 5FIB, 5HQN, 4X5N, 5J36, 5J37, and many others).

3- The fact that the solvent' B-factor (74) is smaller than that of the protein (100) is weird. DO the authors mean protein for complex? In structure PI3kg-PIKiN2.

AU-3: The overall B-factor of the entire PI3KiN2 complex (protein, ligands and solvent) is ~100. We agree that the term “complex” is an awkward term, and we have quoted B-values for protein, inhibitor, and solvent. The B-factor values within the protein vary from 40-220. The interior of the protein has smaller B-factor values, with the exterior of the complex having higher B-factors (see Figure 5.1 below for the reviewers, with the two structures colored by B-factor values). The ordered solvent and inhibitor are bound primarily within the protein’s interior, and have B values very close to the surrounding protein residues. Regions on the exterior of the protein had no ordered solvent. Solvent and inhibitor B-factors are only of a major concern if they are different from the B-factors of surrounding protein residues, and the attached figures verify that indeed bound ligands or solvent

have very similar B-values.

Figure 5.1 for reviewers. The structures for PIKiN2 and PIKiN3 bound to PI3K γ are shown above, colored according to the B-factor for protein, inhibitor, and solvent using PyMol according to the heat map on the right. Solvent is shown as spheres, the inhibitors are shown as sticks, and protein is shown in cartoon representation.

4- The fact that the solvent and inhibitor have smaller b-factor (by a lot) than the 'complex' in such a large structure is concerning. This should be fixed and explained.

AU-4: For PI3K structures, please see answer from the previous question.

In all the T2R-TTL complex structures the chains A, B and C feature lower average B-factors compared to chains D, E and F. The secondary structural elements exposed to the solvent feature higher B-factors compared to the core elements. In all the four complexes the B-factors of the ligands have similar values to their surrounding residues in their binding sites (see Fig. 5.2. for reviewers).

Figure 5.2. for reviewers: The four T2R-TTL structures in complex with BKM120, MTD147, MTD265-

R1 and MTD265 are colored according to B-factors for protein, inhibitor, and solvent in a blue-white-red spectrum (minimum=20; maximum=180). The solvent and ligand molecules are shown as spheres, the protein molecules are in ribbon representation.

5- The complexes of Tubulin with all 4 inhibitors have target of bond length and angles that do not deviate enough from ideality (0.008). Please, re refine with more realistic targets

AU-5: We re-refined all the four structures in PHENIX with weight optimization for geometry and B-factor restraints. All the four re-refined structures yielded lower rmsd values compared to refinement without weight optimization. Refinement performed by loosening the X-ray stereochemistry weight in PHENIX (wxc scale 1.5) yielded rmsd values closer to the ones reported by Engh & Huber, but higher Rfree values, more rotamer outliers and worse Ramachandran statistics. We therefore choose to keep the weight optimization for the final refinement in PHENIX.

6- It is absolutely require that the PDB ID codes are included in the manuscript before it is accepted for publication.

AU-6: The PDB submission for PI3K bound to PIKiN2 and PIKiN3 have been deposited (PDB IDs: 5JHA and 5JHB), and the PDB validation reports are included with the revised manuscript. The complexes of the T2R-TTL complex with all four inhibitors have been deposited with the PDB IDs 5M7E, 5M7G, 5M8D and 5M8G. The validation reports are also included with the revised manuscript.

7- The biological experiments are reported as results of 3 independent experiments as usually required.

AU-7: we rechecked that n's are always clearly noted.

8- Typos and minor correction

AU-7: done.

9- Supplementary Table 6 (is now S8)

AU-9: Minor corrections have been made in Supplementary Table 6 (now 8) to match the style of the new Supplementary Tables 7 and 8.

10- Add commas between cell dimensions to match style of Table 5 (now 7)

AU-10: done (now Supplementary Tables 7 and 8).

Reviewers' Comments:

Reviewer #1 (Remarks to the Author):

The authors have successfully addressed all the reviewer comments.

Reviewer #2 (Remarks to the Author):

The manuscript has been improved.

Reviewer #3 (Remarks to the Author):

Re-review Comments:

This reviewer has reviewed the revised manuscript, portions of the revised supporting information file, and portions of the response to reviews document. The authors have done a fine job in responding to reviewer comments, revising the manuscript, and adding additional experimental data and details as necessary, thus the manuscript should now be suitable for publication. The authors may wish to consider the following additional points in the final version:

1) Abstract: Is it just one atom change or actually a nitrogen atom for a methine group (CH)? This should be clarified in the abstract.

2) If one simply reads the two sentences below without analyzing the actual data, it could lead to a bit of confusion. In other words, since BKM120 demonstrates activity as both an inhibitor of tubulin assembly and as a PI3K inhibitor, then how does one reach the conclusion (at this point in the manuscript) that tubulin activity accounts for the majority of its activity in this head to head cell-based growth inhibition experiment in comparison with PQR309, which has PI3K activity but no tubulin activity? I guess the authors' point is that if both modes of action for BKM120 were equal contributors, then it would show more pronounced GI50 at the same concentration as PQR309, or is this reviewer confused? The sentence might benefit from a bit of further clarification. The two sentences are quoted: "Although BKM120 and PQR309 showed a different mode of action in the above assays, both achieved half-maximal growth inhibition at indistinguishable concentrations at approximately 1 μ M (Supplementary Fig. 1d; Supplementary Fig. 1i-m). This result indicates that the MDA activity of BKM120 dominates its biological action." This reviewer notes that further explanation is provided in the paragraph on Clinical Concentrations.

3) Did the authors model PQR309 with tubulin and confirm that molecular modeling also supports a non-interaction with tubulin? This might be useful in order to further cement the differences between PQR309 (non-tubulin active) and BKM120 and MTD147. Apologies in advance if these data are in the manuscript but missed by this reviewer.

4) The modeling experiments suggest an interaction at the colchicine site. Therefore, why not do a competitive binding assay with radiolabeled colchicine to confirm this experimentally? The authors respond to this in their response to reviews, but it is still a bit confusing to this reviewer why the assay is not carried out. This relates to point #5 (below). This is not a deal breaker in terms of moving forward with publication.

5) Unless the authors have any direct experimental binding data, the following sentences should be modified: “Despite its structural similarity to BKM120 and MTD147, PQR309 does not bind to tubulin.” “This conclusion readily explains why PQR309 and the BKM120 regioisomer (BKM120-R1, Supplementary Fig. 4 g-j) do not bind to tubulin, as there is no possible orientation to prevent repulsive forces between their core nitrogens and β Met259.”

6) Line 266, consider “...therapeutically relevant...”

7) Consider revising the following sentence to stress the change from N to CH rather than emphasizing one Dalton. “We succeeded in separating the activities of BKM120 as a PI3K inhibitor and an MDA by a one Dalton chemical modification of its core pyrimidine ring.”

8) A quick review of the added data for inhibition of tubulin polymerization (cell-free assay) in Supp Figs 2 and 4 and the IC₅₀ values in Supp Table 4, is interesting. If one looked only at Supp Table 4 and skipped over the rest of the manuscript then it would not seem that inhibition of tubulin polymerization is the main mode of action for BKM120 since its IC₅₀ value was 23 micromolar (modest at best) in comparison to nocodazole at 2.7 micromolar. The cut-off value (for no activity) in this assay appeared to be >45 micromolar. MTD147 and MTD265 were strong inhibitors (3.7 and 2.1 micromolar, respectively). Obviously this is only one portion of data and when one looks at the compilation of overall data in the manuscript, then the overall conclusions drawn by the authors are supported. Still, it remains interesting to this reviewer that BKM120 has only modest activity as an inhibitor of tubulin polymerization in this cell-free (pure protein) assay.

Reviewer #4 (Remarks to the Author):

Now it is well refined and suitable for publication.

Reviewer #5 (Remarks to the Author):

The authors have addressed all the comments that I've raised. For example, they've now added that they have refined the structure with Phenix. This detail justifies and corroborates the bond length obtained.

2) Revision of revised article

Reviewer #1 (Remarks to the Author AU):

The authors have successfully addressed all the reviewer comments.

Reviewer #2 (Remarks to the Author AU):

The manuscript has been improved.

Reviewer #3 (Remarks to the Author AU):

Re-review Comments: This reviewer has reviewed the revised manuscript, portions of the revised supporting information file, and portions of the response to reviews document. The authors have done a fine job in responding to reviewer comments, revising the manuscript, and adding additional experimental data and details as necessary, thus the manuscript should now be suitable for publication. The authors may wish to consider the following additional points in the final version:

1 - Abstract: Is it just one atom change or actually a nitrogen atom for a methine group (CH)? This should be clarified in the abstract.

AU-1: The reviewer's comment is formally correct. To non-chemists, the "one atom change" (N vs C) is actually more meaningful, and simpler to convey. We would therefore like to keep the present text.

2 - If one simply reads the two sentences below without analyzing the actual data, it could lead to a bit of confusion. In other words, since BKM120 demonstrates activity as both an inhibitor of tubulin assembly and as a PI3K inhibitor, then how does one reach the conclusion (at this point in the manuscript) that tubulin activity accounts for the majority of its activity in this head to head cell-based growth inhibition experiment in comparison with PQR309, which has PI3K activity but no tubulin activity? I guess the authors' point is that if both modes of action for BKM120 were equal contributors, then it would show more pronounced GI50 at the same concentration as PQR309, or is this reviewer confused? The sentence might benefit from a bit of further clarification. The two sentences are quoted:

"Although BKM120 and PQR309 showed a different mode of action in the above assays, both achieved half-maximal growth inhibition at indistinguishable concentrations at approximately 1 μ M (Supplementary Fig. 1d; Supplementary Fig. 1i-m). This result indicates that the MDA activity of BKM120 dominates its biological action."

This reviewer notes that further explanation is provided in the paragraph on Clinical Concentrations.

AU-2: We have changed the text to make the sentences easier to read, and to make clear that the last sentence refers not only to the cited text, but to both paragraphs preceding this [preliminary] conclusion. As the reviewer points out, additional results and discussion covering this statement follow later.

3 - Did the authors model PQR309 with tubulin and confirm that molecular modeling also supports a non-interaction with tubulin? This might be useful in order to further cement the differences between PQR309 (non-tubulin active) and BKM120 and MTD147. Apologies in advance if these data are in the manuscript but missed by this reviewer.

AU-3: There is sufficient data presented to illustrate that PQR309 does not interact with tubulin: PQR309 could not be soaked into tubulin crystals, does not lead to cellular pHistone H3 increase, mitotic arrest, and did not block *in vitro* microtubule polymerization or microtubule dynamics. The argumentation based on BKM120, BKM120-R1, MTD147, MTD265 and MTD265-R1 X-Ray structures and biochemical and cellular data provides a sound argumentation that we feel has more weight than modeling. Although possible to add, a modeling section would to our opinion rather dilute than enhance the flow and conclusion of the manuscript.

4 - The modeling experiments suggest an interaction at the colchicine site. Therefore, why not do a competitive binding assay with radiolabeled colchicine to confirm this experimentally? The authors respond to this in their response to reviews, but it is still a bit confusing to this reviewer why the assay is not carried out. This relates to point #5 (below). This is not a deal breaker in terms of moving forward with publication.

AU-4: We would like to refer to the response AU-10b above. We do not believe that this experiment would deliver the expected results.

5 - Unless the authors have any direct experimental binding data, the following sentences should be modified: "Despite its structural similarity to BKM120 and MTD147, PQR309 does not bind to tubulin." "This conclusion readily explains why PQR309 and the BKM120 regioisomer (BKM120-R1, Supplementary Fig. 4 g-j) do not bind to tubulin, as there is no possible orientation to prevent repulsive forces between their core nitrogens and β Met259."

AU-5: It is a fact, that at reasonable concentrations PQR309 does not show any signs of microtubule interaction (no pHis H3, no mitotic arrest, no effect on MT dynamics, no prevention of MT polymerisation, etc.). Moreover, soaking of tubulin crystals with PQR309 was not successful. Additionally, the effect of the core nitrogen orientation has been documented with the analysis of MTD265 and MTD265-R1 (and compounds not shown here). The text as provided above is to our knowledge currently the best way to explain why PQR309 does not bind to tubulin. Scientifically, the absence of any event (here binding) can be formally not proven...

6 - Line 266, consider "...therapeutically relevant..."

AU-6: Done.

7 - Consider revising the following sentence to stress the change from N to CH rather than emphasizing one Dalton. "We succeeded in separating the activities of BKM120 as a PI3K inhibitor and an MDA by a one Dalton chemical modification of its core pyrimidine ring."

AU-7: We have replaced "a one Dalton" by "a minimal".

8 - A quick review of the added data for inhibition of tubulin polymerization (cell-free assay) in Supp Figs 2 and 4 and the IC₅₀ values in Supp Table 4, is interesting. If one looked only at Supp Table 4 and skipped over the rest of the manuscript then it would not seem that inhibition of tubulin polymerization is the main mode of action for BKM120 since its IC₅₀ value was 23 micromolar (modest at best) in comparison to nocodazole at 2.7 micromolar. The cut-off value (for no activity) in this assay appeared to be >45 micromolar. MTD147 and MTD265 were strong inhibitors (3.7 and 2.1 micromolar, respectively). Obviously this is only one portion of data and when one looks at the compilation of overall data in the manuscript, then the overall conclusions drawn by the authors are supported. Still, it remains interesting to this reviewer that BKM120 has only modest activity as an inhibitor of tubulin polymerization in this cell-free (pure protein) assay.

AU-8: As we pointed out above (several answers in round one), the turbidity assay for microtubule polymerization has some inherent serious limitations (as [tubulin] and [drug] must be very high). This is a reason why we provided initially only the data covering microtubule dynamics measured by GFP-EB3 assays, which deliver a number of important output parameters, and show a good agreement with cellular data for BKM120 (effective at < 1 μ M). In the table legend of Supplementary Table 4 we point to these limitations: "***The dynamic range of the turbidity-based microtubule polymerization assay is limited by the tubulin vs. compound stoichiometry, which sets the lowest IC₅₀ detection limit to around 2-3 μ M." While the turbidity assay for tubulin polymerization is well suited to determine if a compound interacts with tubulin, IC₅₀ values have to be interpreted with care. We agree with the reviewer that BKM120 is not a very high affinity tubulin binder, as demonstrated with MTD147 and MTD265 derivatives, but its tubulin interaction is sufficiently high to dominate its effects as a PI3K inhibitor.

Reviewer #4 (Remarks to the Author AU):

Now it is well refined and suitable for publication.

Reviewer #5 (Remarks to the Author AU):

The authors have addressed all the comments that I've raised. For example, they've now added that they have refined the structure with Phenix. This detail justifies and corroborates the bond length obtained.

AU-to all reviewers: We would like to thank all reviewers and editors for their critical and constructive comments that have improved the manuscript considerably.